# Deep brain stimulation of thalamic nucleus reuniens promotes neuronal and cognitive resilience in an Alzheimer's disease mouse model

Shiri Shoob[1], Nadav Buchbinder[1,2,4], Ortal Shinikamin[1,4], Or Gold[3], Halit Baeloha[1], Tomer Langberg[1,2], Daniel Zarhin[1], Ilana Shapira[1], Gabriella Braun[1,2], Naomi Habib [3] & Inna Slutsky [1,2] ✉

The mechanisms that confer cognitive resilience to Alzheimer's Disease (AD) are not fully understood. Here, we describe a neural circuit mechanism underlying this resilience in a familial AD mouse model. In the prodromal disease stage, interictal epileptiform spikes (IESs) emerge during anesthesia in the CA1 and mPFC regions, leading to working memory disruptions. These IESs are driven by inputs from the thalamic nucleus reuniens (nRE). Indeed, tonic deep brain stimulation of the nRE (tDBS-nRE) effectively suppresses IESs and restores firing rate homeostasis under anesthesia, preventing further impairments in nRE-CA1 synaptic facilitation and working memory. Notably, applying tDBS-nRE during the prodromal phase in young APP/PS1 mice mitigates age-dependent memory decline. The IES rate during anesthesia in young APP/PS1 mice correlates with later working memory impairments. These findings highlight the nRE as a central hub of functional resilience and underscore the clinical promise of DBS in conferring resilience to AD pathology by restoring circuit-level homeostasis.

Alzheimer's disease (AD) is a progressive neurodegenerative disorder that accounts for the vast majority of dementia cases. Pathologically, it is characterized by amyloid deposits and neurofibrillary tangles present in the brain upon autopsy. Clinically, it manifests as a progressive decline in cognitive functions. The presymptomatic phase of the disease, occurring 10–20 years before cognitive impairments, is marked by early amyloid-β deposition, followed by cortical hypometabolism, accumulation of tau pathology and hippocampal volume loss[1,2]. Moreover, ~30% of individuals with AD-type neuropathology do not meet criteria for dementia during their lifetime[3]. Despite significant progress in understanding the biochemical[4], synaptic[5] and network-wide[6] abnormalities associated with AD, the circuit mechanisms that

underlie resilience to synaptic, circuit, and cognitive dysfunctions in the face of neuropathology remain largely unknown[7]. Gaining insights into the neurobiological basis of stable, resilient states may provide strategies to prevent the transition from dormant to symptomatic disease phase.

Aberrations in low-arousal brain states, such as those occurring during sleep[8–12] and induced by general anesthesia[13–15], have been associated with an increased risk of AD pathology and cognitive impairments. Extensive research has examined the bidirectional relationships between sleep and AD-related pathology[12,16–22]. Nevertheless, the mechanisms by which another low-arousal brain state, induced by general anesthesia, impacts the progression of the disease, remain

[1]Department of Physiology and Pharmacology, Faculty of Medicine, Tel Aviv University, 69978 Tel Aviv, Israel. [2]Sagol School of Neuroscience, Tel Aviv University, 69978 Tel Aviv, Israel. [3]Edmond & Lily Safra Center for Brain Sciences, The Hebrew University of Jerusalem, Jerusalem, Israel. [4]These authors contributed equally: Nadav Buchbinder, Ortal Shinikamin. ✉e-mail: islutsky@tauex.tau.ac.il

elusive. Postoperative cognitive dysfunction (POCD) is a common occurrence, particularly among the elderly, and its symptoms may persist for months or even years[23]. Studies in familial AD (fAD) mouse models have predominantly focused on investigating the role of amyloid and tau pathologies in anesthesia-induced cognitive impairments[14,23–25]. Our recent work reveals that general anesthesia exposes dyshomeostasis of CA1 neuronal activity and interictal epileptiform spikes (IESs) during the prodromal disease stage, preceding cognitive decline in distinct fAD mouse models[26]. However, some central questions have remained open. What neural circuitry is responsible for the increased susceptibility to cognitive dysfunctions induced by anesthesia in the context of AD? Could the disruption of neuronal activity homeostasis, exposed by anesthesia, accelerate cognitive dysfunctions during the prodromal stage of the disease? Lastly, do the circuit-level mechanisms exposed by anesthesia during the prodromal phase of AD contribute to the subsequent age-dependent cognitive decline as the disease advances?

In this study, we investigated the functional consequences of IESs, and the neuronal circuitry involved in their regulation. Specifically, we focused on investigating the outputs of the nucleus reuniens (nRE) of the midline thalamus, as it serves as the primary source of thalamic inputs to the hippocampal CA1 region[27,28]. We examined whether the nRE plays a role in regulation of IESs and whether this early hyperexcitability contributes to spatial working memory (SWM) deficits in the APP/PS1 fAD mouse model. Our study identifies the crucial role of the nRE in regulating anesthesia-induced IESs in the CA1 and mPFC brain regions during the prodromal disease stage and their role in SWM dysfunctions in the prodromal disease stage. It prompted further investigation into whether the same circuit-level mechanisms are involved in cognitive dysfunctions as the disease advances. Indeed, we found that the rate of IESs predicts the severity of working memory impairment in the later, symptomatic disease stage. Our findings demonstrate the effectiveness of tonic DBS of the nRE (tDBS-nRE) in preventing dysfunctions at the synaptic, network, and cognitive levels. As a result, the nRE-CA1-mPFC circuit emerges as a brain circuit that promotes neuronal and cognitive resilience to amyloid pathology in the context of AD.

## Results

### Anesthesia induces abnormal response at electrophysiological, behavioral, and transcriptional levels in the hippocampus of fAD mice

Recordings of local field potentials (LFPs) in the CA1 area of the hippocampus in early-stage, 4–5 months old APP/PS1 (APP$_{SWE}$/PS1ΔE9) mice revealed frequent high-voltage IESs (3.22 ± 0.51 per min) under general anesthesia (1.5% isoflurane, Fig. 1a-b). IESs rarely happened in wild-type (WT) mice under anesthesia (0.216 ± 0.095 per min, Fig. 1a-b). This indicates that anesthesia exposes early dysfunctions in CA1 area of the hippocampus of APP/PS1 mice, confirming the results of our previous study[26]. CA1 activity is normal during active wakefulness in cognitively unimpaired APP/PS1 mice[26]. We hypothesized that general anesthesia would also expose initial cognitive deficits in early-stage fAD mice. As working memory is a central component of cognitive function and is particularly susceptible to AD[29,30], we asked whether anesthesia produces a lasting disruption to SWM in APP/PS1 mice before the appearance of innate memory impairments.

To address this question, we tested the effect of general anesthesia on the delta-maze alternation test in 4–5 month old WT vs. APP/PS1 naïve mice. The percentage of correct choices (alternations) was recorded for each daily session. Mice were first trained daily with a 10 s delay between sample and choice until each mouse reached a stable success rate. All WT and APP/PS1 mice reached an asymptotic success rate of ~80% by 8–10 days of training, indicating that they learned the task (Supplementary Fig. 1a). Learning was similar between genotypes ($P = 0.64$). Following the training, SWM was tested at 10, 20, and 60 s

delays. Spatial working memory was not impaired in 4–5 month-old APP/PS1 mice, as success rates were similar between genotypes at each of the delays (Supplementary Fig. 1b). The mice were then anesthetized using isoflurane inhalation (1.5%, 3 h) and performance was subsequently tested in the following days. Following anesthesia, APP/PS1 mice (Fig. 1d), but not WT (Fig. 1c), showed a significant ($P < 0.0001$) decrease in choice accuracy in all three delay times. The impairments in SWM in APP/PS1 mice were transient and there was a gradual return to baseline performance levels over the course of 3 days following the anesthesia (Fig. 1e). These experiments demonstrate that general anesthesia induces CA1 hyperexcitability and spatial working memory impairments in otherwise cognitively normal APP/PS1 mice.

To test if the decline in spatial working memory was linked to CA1 hyperexcitability exposed by anesthesia, we repeated the experiment in APP/PS1 mice implanted with LFP electrodes to record IESs in the hippocampal CA1. The CA1-IES rate during anesthesia strongly correlated with the degree of subsequent working memory impairment (Fig. 1f). Based on these results, we hypothesized that epileptiform activity might serve as the triggering signal leading to maladaptive plasticity, ultimately resulting in working memory deficits.

We next tested whether fAD mutations alter the transcriptional response of CA1 excitatory neurons (CA1-ExNs) to anesthesia, which could potentially shed light on the plasticity mechanisms associated with CA1-ExN dyshomeostasis[26] and memory deficits (Fig. 1e) in the model mice. To address this question, we utilized single nuclei RNA-sequencing (snRNAseq) to profile 5796 nuclei of CA1-ExNs and performed a comparative analysis based on brain state and genotype in 5-month-old mice. Hippocampi were isolated from WT and APP/PS1 littermates after either wake or general anesthesia (GA: 1.5% isoflurane, 3 h), at the same circadian time. Four groups of mice (15 mice, 3–4 mice per group) were analyzed: WT-Awake, WT-GA, fAD-Awake, and fAD-GA. In the awake state, we identified 89 differentially expressed genes (DEGs) between WT and APP/PS1 mice, while in the GA state, there were 119 DEGs (Fig. 1g and Source Data). In both brain states (awake and GA), the DEGs were found to be enriched in pathways related to synapse organization, synaptic signaling, neural development, and neuronal projections. Notably, under anesthesia, additional DEGs between the genotypes were observed, which were related to RNA processing/splicing and cell motility (based on GO biological pathway enrichment analysis, average logFC> 0.2, FDR correction <0.05). Among the top 20 DEGs under anesthesia (Fig. 1h), many are closely associated with synaptic functions, neurodevelopment and neurodevelopmental disorders[31–40] (e.g. Nrg3os, Pigk, Ube3a, Kcnn2, Lrrc4, Fez1, Rsrp1, Aplp1, Ogt, Lin7b). We also identified DEGs related to Aβ / tau proteostasis[41–43] (Rtn1, Dnaja2, Srsf7), sleep abnormalities[44,45] (Gnas, Ube3a), and neurodegeneration[46,47] (Ogt, Ssbp4). As our anesthesia regime did not result in detectable changes in the levels of soluble Aβ40 and Aβ42 peptides or their ratio (Supplementary Fig. 2), it is unlikely that the observed transcriptional changes are downstream of Aβ.

Altogether, our multi-level analysis uncovers the distinct response of hippocampal circuits to anesthesia at the transcriptional, electrophysiological, and behavior levels in fAD mice.

### Anesthesia impairs nRE-CA1 short-term synaptic plasticity in fAD mice

Working memory function relies on the interconnected circuitry of the hippocampus, medial prefrontal cortex (mPFC), and thalamic nucleus reuniens[48–51] (nRE). The nRE, situated in the midline limbic thalamus, serves as the principal thalamic input to the hippocampal CA1 region[52]. Additionally, the thalamus plays a crucial role in natural arousal regulation and becomes deactivated during anesthesia[53–55]. Building on this knowledge, we hypothesized that dysregulation of nRE-CA1 synaptic connections might be responsible for anesthesia-induced hyperexcitability and working memory impairments in fAD mice.

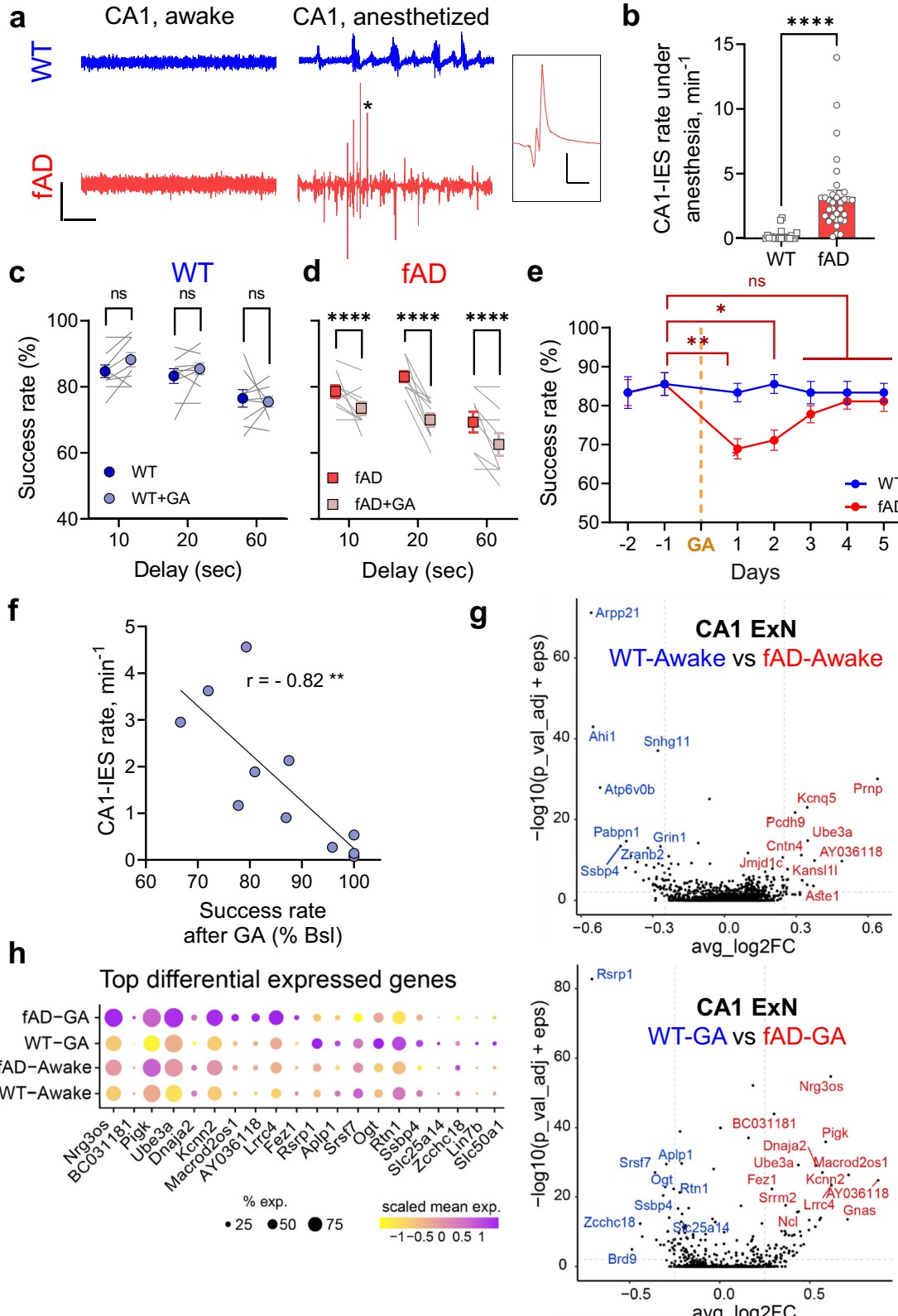

To test this, we implanted a stimulating electrode into the nRE and recorded field postsynaptic potentials (fEPSPs) in the CA1. Short-term synaptic facilitation is considered essential for maintaining working memory[56]. We quantified the effect of anesthesia on short-term synaptic plasticity evoked by high-frequency burst stimulation at the nRE-CA1 synaptic connections in awake APP/PS1 mice, in addition to basal synaptic transmission during low-frequency stimulation. Measurements were taken at the same circadian time. Short-term synaptic facilitation at the nRE-CA1 synapse was normal in APP/PS1 mice in the baseline (paired-pulse ratio 1.93 ± 0.12 in WT and 2.01 ± 0.1 in APP/PS1, Fig. 2a, b and Supplementary Fig. 1d). Anesthesia led to a profound reduction in paired-pulse ratio only in APP/PS1 mice (to 1.24 ± 0.18), which remained low for 3 days before returning to baseline levels (Fig. 2a, b). This suggests increased release probability in nRE-CA1 synapses of APP/PS1 mice

**Fig. 1 | Anesthesia induces CA1 hyperexcitability, transient impairment of spatial working memory, and abnormal transcriptional response of CA1 excitatory neurons in fAD, but not in WT mice. a** Representative CA1 LFP traces in WT (top) and fAD (bottom) mice during awake (left) and general anesthesia (GA: 1.5% isoflurane, right). fAD mice show interictal epileptiform spikes (IESs) under anesthesia. Scale bars: 10 s, 2 mV. Zoom-in on IES marked by an asterisk. Scale bars: 50 ms, 2 mV. **b** fAD mice had more IESs during GA (WT $n = 22$, fAD $n = 32$, $P < 0.0001$, Mann–Whitney two-tailed test). **c** Spatial working memory (SWM) was stable in WTs ($n = 9$) after GA ($P = 0.34$, Two-way ANOVA, Sidak's multiple comparisons: $P = 0.27$, 0.65 and 0.94 for 10, 20 and 60 sec delays, respectively). **d** SWM was impaired in fAD mice ($n = 9$) following GA ($P = 0.0046$, Two-way ANOVA, Sidak's multiple comparison tests: $P < 0.0001$ for all the delays). **e** Delta maze success rate of WT (blue, $n = 9$) and fAD (red, $n = 9$) male mice before and after GA for the 20 s delay. SWM impairments were transient in fAD group ($P = 0.043$, 0.027,

and >0.9 for day 1, 2, and 3–5 days after GA, respectively; Friedman test, Dann's multiple comparisons). **f** CA1-IES frequency inversely correlated with post-GA SWM (Spearman $r = -0.9154$, $P = 0.0016$, $n = 11$). **g** Differentially expressed genes in fAD vs. WT male mice during awake (top) and GA (bottom) states in CA1-ExN. Volcano plots depict log-fold-change (FC, x-axis) and -log(P-value) (y-axis) of differential gene expression. snRNA-seq data from 4 mice per group (WT-Awake, fAD-Awake, and WT-GA), and 3 mice in fAD-GA group. 2425 and 2807 CA1-ExN nuclei in awake, 2497 and 3299 CA1-ExN nuclei in GA, for fAD and WT, respectively; MAST test $P$ value, FDR correction. FC, log average expression fold change. **h** Expression patterns of top up- and down-regulated genes across genotypes and states within CA1-ExN nuclei profiled by snRNA-seq. Dot color represents column-scaled expression level of the expressing cells in the group, and size is the percentage of cells in the cell type expressing the gene. Error bars represent SEM; ns – non-significant, *$P < 0.05$, **$P < 0.01$, ***$P < 0.001$, ****$P < 0.0001$.

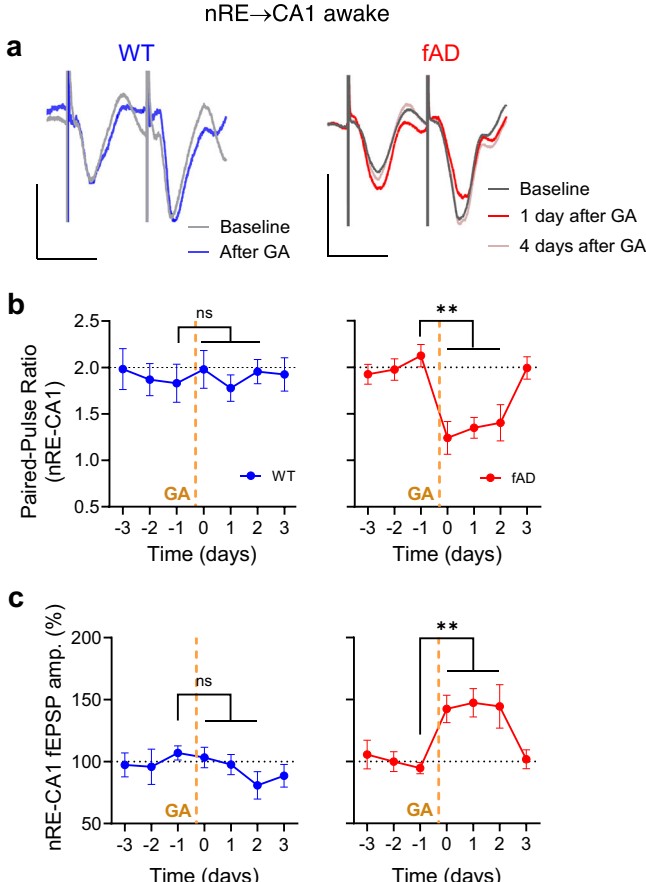

**Fig. 2 | Anesthesia transiently augments synaptic transmission and decreases nRE-CA1 short-term synaptic facilitation in fAD, but not in WT mice.**
**a** Representative fEPSP traces evoked by 0.1 mA stimulation at 25 Hz in nRE-CA1 synapses before and after GA in awake WT and fAD mice. Scale bars: 30 ms, 0.2 mV (WT); 30 ms, 0.5 mV (fAD). **b** Dynamics of paired-pulse facilitation before and after anesthesia in WT (left, $n = 6$; 4 males, 2 females) and fAD (right, $n = 8$; 5 males, 3 females) mice. Friedman test with Dann's multiple comparison tests: $P = 0.0012$, 0.0005, and 0.0078 (fAD) and $P > 0.5$ (WT) for the first, second, and third day after GA in comparison to the day before GA. **c** The dynamic change in the amplitude of fEPSP evoked by the first stimuli of the 25 Hz burst (at 0.03 Hz) normalized to baseline, before and after anesthesia in WT (left, $n = 6$; 4 males, 2 females) and fAD (right, $n = 8$; 5 males, 3 females) mice. Friedman test with Dann's multiple comparison tests: $P = 0.0047$, 0.002, and 0.0072 (fAD) and $P > 0.8$ (WT) for the first, second, and third day after GA in comparison to the day before GA. Error bars represent SEM; ns non-significant, **$P < 0.01$.

since basal synaptic responses to low-frequency stimuli were enhanced for 2 days following anesthesia in APP/PS1 but not in WT mice (Fig. 2c). In summary, general anesthesia led to impairments in nRE-CA1 short-term synaptic facilitation and spatial working memory in APP/PS1 mice, which persisted for two days after anesthesia.

## Inhibiting the nRE-CA1 pathway suppresses CA1 hyperexcitability

We next tested whether CA1 hyperexcitability observed in anesthetized APP/PS1 mice was due to synaptic inputs originating from the nRE. Indeed, our initial findings revealed a notable increase in the number of neurons expressing the activity-regulated immediate early gene *cfos* in the midline thalamus of APP/PS1 mice after anesthesia, in comparison to WT mice (Fig. 3a, b). Next, we tested the impact of silencing the nRE on CA1 hyperexcitability during anesthesia. Pharmacological silencing of spikes in the nRE area of anesthetized APP/PS1 mice by local injection of tetrodotoxin (TTX) caused a significant reduction in the CA1-IES rate compared to baseline (Fig. 3c and Supplementary Fig. 3a, b). Blocking excitatory synaptic transmission in the nRE by locally injecting antagonists of AMPA and NMDA receptors (CNQX and AP5) led to a similar suppressive effect on CA1 hyperexcitability (Fig. 3d). In contrast, injecting a vehicle (control) did not cause a significance change in the CA1-IES rate (Supplementary Fig. 3c).

To directly test whether inhibiting the nRE-CA1 synapse can suppress anesthesia-induced CA1-IESs in fAD model mice, we combined a chemogenetic designer receptors exclusively activated by designer drugs (DREADD) approach[57] and a retrograde adeno-associated virus[58] to specifically target nRE-CA1 synapses. We injected AAV2retro-CaMKIIα-iCre in the CA1 of APP/PS1 mice, along with Cre-dependent hM4D(Gi)-mCherry virus in the nRE. This allowed us to express the inhibitory DREADD hM4D(Gi)-mCherry in excitatory nRE neurons, forming a monosynaptic connection to the CA1 (Fig. 3e). As expected, the DREADD agonist CNO effectively suppressed nRE-CA1 synaptic transmission (Supplementary Fig. 3d, e). To quantitatively assess the impact of inactivating nRE-CA1 synapses on CA1 hyperexcitability, we measured the effect of CNO on the IES rate during anesthesia. Intraperitoneal (i.p.) injections of CNO led to an average ~47% decrease in the rate of CA1 IESs compared to the same volume of vehicle (VEH) injections per mouse (Fig. 3f and Supplementary Fig. 3f, g). Importantly, CNO did not affect the IES rate in a control group of mice, expressing Cre-dependent mCherry in the nRE and AAV2retro-CaMKIIα-iCre in the CA1 (Supplementary Fig. 3h, i).

These results demonstrate that the nRE-CA1 pathway can regulate, and most likely contributes to, anesthesia-induced CA1 hyperexcitability in APP/PS1 mice.

## Phasic and tonic stimulation of the nRE bi-directionally modulates CA1 epileptiform activity

DBS has shown clinical benefits in Parkinson's disease and is being considered as a promising therapy for AD[59]. Therefore, we

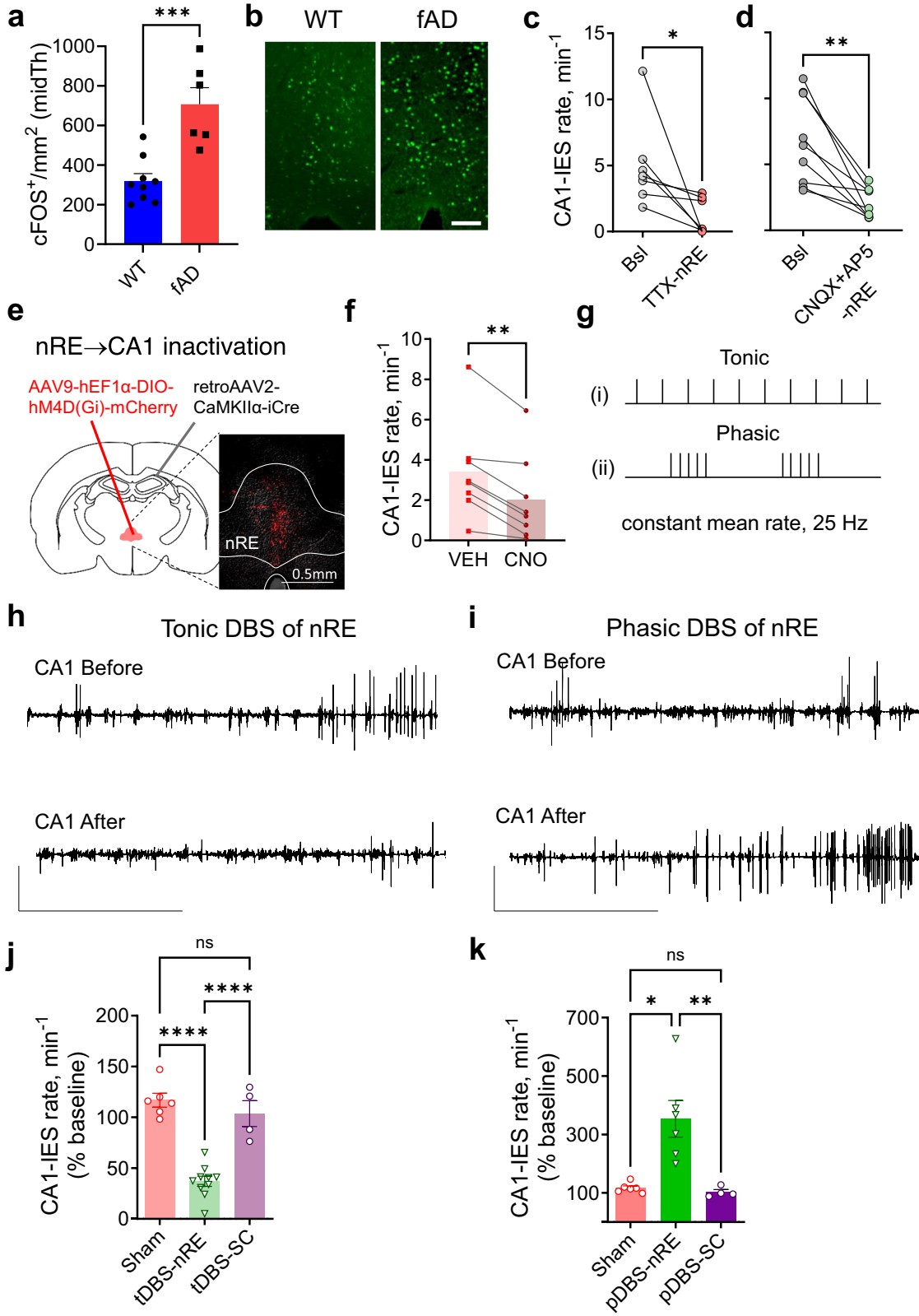

decided to explore the potential of DBS to modulate CA1 hyperexcitability in APP/PS1 mice. Considering the role of the thalamus as a 'choke point' in epileptic circuits[60] and its ability to abort absence seizures by switching thalamocortical neurons from a phasic to tonic firing[61], we investigated how tonic vs. phasic stimulation of the nRE modulates CA1 hyperexcitability in APP/PS1 mice.

We implanted a stimulating electrode in the nRE and a recording electrode in the CA1 of APP/PS1 mice and designed two DBS protocols that had the same duration and number of stimuli, but different temporal patterns. The first protocol, termed 'tonic DBS' of nRE (tDBS-nRE), involved constant stimulation at 25 Hz for 400 s (0.2 mA amplitude, 0.5 ms pulse duration). The second protocol, termed 'phasic DBS' (pDBS-nRE), delivered bursts of 125 stimuli at 100 Hz, with 3.75-s

**Fig. 3 | Bi-directional regulation of CA1 hyperexcitability by tonic versus phasic nRE stimulation under anesthesia in fAD model. a, b** Higher number of c-Fos+ cell count in midline thalamus post-anesthesia in fAD mice ($n = 6$) compared to WT ($n = 9$) ($P = 0.0008$, Mann–Whitney two-tailed test). Scale: 100 μm. **c** Local TTX injection (300 nl, 70 μM) to nRE reduced CA1-IES rate in anesthetized fAD mice ($P = 0.0156$, Wilcoxon two-tailed test, $n = 7$; 3 males, 4 females). **d** Local CNQX + AP5 injection (1 μl, CNQX 5 mM + AP5 25 mM) to nRE decreased CA1-IES rate in fAD mice ($P = 0.0039$, Wilcoxon two-tailed test, $n = 9$). **e** Mice received retrograde viral vector carrying iCre in CA1 SLM and viral vector carrying Cre-dependent hM4D(Gi)-mCherry in the nRE, expressing hM4D(Gi)-mCherry specifically in nRE cells projecting to CA1. Scale bar: 0.5 mm. **f** CA1-IES rate after VEH (saline) vs. CNO i.p. injections (5 mg/kg) in fAD mice with hM4D(Gi)-mCherry in nRE-CA1 synapses ($P = 0.0078$, Wilcoxon two-tailed test, $n = 8$ mice; 6 males, 2 females). **g** Stimulation patterns: nRE and SC stimulated for 400 seconds - (i) tonic (tDBS) or (ii) phasic: 1.25 s 100 Hz bursts every 5 s (pDBS). **h, i** Representative CA1 IESs traces in baseline and after tDBS (**h**) or pDBS (**i**) to nRE in fAD mice. Scale bars: 5 mV, 1 min. **j** fAD mice had lower CA1-IES frequency during GA a week after tDBS in nRE, but not SC ($P < 0.0001$, One-way ANOVA with Sidak's multiple comparisons: Sham vs nRE and nRE vs SC $P < 0.0001$, $n = 6, 10$, and 4 for Sham, nRE, and SC, male / female ratio: 4/2, 7/3, 3/1, respectively). **k** fAD mice exhibited higher CA1-IES frequency during GA a week after pDBS in nRE, but not SC ($P = 0.0002$, Kruskal–Wallis test with Dunn's multiple comparisons: Sham vs nRE $P = 0.0326$, nRE vs SC $P = 0.0060$, $n = 6, 6$, and 4 for Sham, nRE, and SC, male / female ratio: 4/2, 4/2, 3/1, respectively). All the experiments are performed in anesthetized mice. Error bars represent SEM; ns – non-significant, $*P < 0.05$, $**P < 0.01$, $****P < 0.0001$.

inter-burst interval during the same 400-s period (Fig. 3g). As controls, we used a sham condition (nRE electrode without stimulation) and stimulation of the Schaffer Collateral (SC) pathway, which is the major excitatory input to the CA1[62]. All DBS protocols were delivered during general anesthesia.

The results showed that tDBS-nRE led to a significant ~60% decrease in the CA1-IES rate compared to baseline, persisting for at least a week after the stimulation (Fig. 3h, j and Supplementary Fig. 4a). In contrast, pDBS-nRE resulted in a notable ~250% increase in the rate of IESs (Fig. 3i, k and Supplementary Fig. 4b). Importantly, SC pathway stimulation had no significant effect on CA1 epileptiform activity using the same tonic and phasic stimulation protocols (Fig. 3j, k and Supplementary Fig. 4c). Additionally, we observed similar effects of tDBS-nRE on CA1 IESs when using a different anesthetic (ketamine-xylazine, Supplementary Fig. 5). These findings indicate that the impact of tDBS-nRE is not specific to isoflurane-induced hyperexcitability. Given that ketamine impairs working memory function irrespective of IESs even in wild-type rodents[63,64] via NMDA receptor blockade[65], we did not continue investigating its effects on working memory in fAD mice.

### Phasic and tonic stimulation of the nRE bi-directionally modulates mPFC epileptiform activity

The nRE acts as the main connection between hippocampal CA1 and the mPFC[28] (Fig. 4a). Therefore, we hypothesized that anesthesia might also trigger hyperexcitability in the mPFC through the nRE. We performed simultaneous recordings in both CA1 and mPFC in anesthetized APP/PS1 mice and indeed observed IESs in the mPFC, similar to those recorded in the CA1 (Fig. 4b). Most CA1 IESs were isolated in time from mPFC IESs (Fig. 4c). Despite this, CA1 and mPFC IESs were significantly cross-correlated at the scale of milliseconds (Fig. 4d, e, f). Mice with higher IES rate in CA1 also had higher IES rate in mPFC, recorded during the same isoflurane session (Fig. 4g). Similarly to epileptiform activity in the CA1 (Fig. 1f), the mPFC-IES rate during anesthesia was negatively correlated to spatial working memory (Fig. 4h).

We further quantified the effects of tonic vs. phasic DBS-nRE on the rate of IES in the mPFC of APP/PS1 mice. Similar to our findings in CA1, tonic DBS-nRE suppressed, while phasic DBS-nRE exacerbated, mPFC hyperexcitability (Fig. 4i and Supplementary Fig. 6a–c). These results demonstrate the bidirectional regulatory role of nRE on CA1 and mPFC hyperexcitability, highlighting the potential of tonic DBS in treating AD-related hyperexcitability.

### tDBS-nRE restores anesthesia-induced suppression of CA1 activity

So far, we have shown that tDBS-nRE is effective in preventing anesthesia-induced CA1 hyperexcitability, as indicated by the suppression of epileptiform activity. In light of our recent findings, which demonstrated a loss of negative regulation of CA1 firing rates in APP/PS1 mice during general anesthesia[26], we sought to investigate whether tDBS-nRE could also effectively restore the normal regulation of CA1

firing rates during anesthesia. To answer this question, we adapted a wide-field, head-mounted miniaturized microscope[26,66–68] to track Ca²⁺ dynamics as a proxy for neuronal spiking activity in thousands of CA1 neurons with single-cell resolution (Fig. 5a, b). We used a genetically encoded Ca²⁺ sensor GCaMP6f[69] expressed in excitatory CA1 pyramidal neurons under the CaMKIIα promoter. Imaging was performed at regular hours to avoid circadian effects. Each imaging session consisted of awake and anesthetized states in the same freely behaving mice. As reported previously[36], general anesthesia inhibited CA1 population activity in WT mice (Supplementary Fig. 7a), but not in APP/PS1 mice (Fig. 5c, f, g). Indeed, tDBS-nRE enabled the suppression of CA1 activity by anesthesia in APP/PS1 mice one week after stimulation (Fig. 5d, h). The degree of suppression by anesthesia in APP/PS1 mice was comparable to the effect of anesthesia in WT mice (Fig. 5f and Supplementary Fig. 7d, e). Strikingly, this effect of tDBS-nRE remained stable for one month following the stimulation (Fig. 5e, i and Supplementary Fig. 7d). Interestingly, the CA1-IES rate correlated with the effect of anesthesia on CA1 firing rate (Supplementary Fig. 7f). These results demonstrate that tDBS-nRE restores physiological down-regulation of CA1 activity by general anesthesia in APP/PS1 mice and suggest that rescuing state-dependent regulation of CA1 firing rates contributes to the suppression of CA1 hyperexcitability by tDBS-nRE.

### tDBS-nRE prevents anesthesia-induced synaptic and memory dysfunctions

We have shown so far that nRE-CA1 short-term synaptic plasticity is impaired by anesthesia in APP/PS1 mice (Fig. 2a, b) and that the tDBS-nRE restores CA1 firing rates (Fig. 5) and suppresses CA1 and mPFC IESs (Figs. 3–4) during anesthesia in APP/PS1 mice. We next hypothesized that tDBS-nRE can also restore anesthesia-induced synaptic and cognitive impairments. To test this hypothesis, we implanted APP/PS1 mice with stimulating electrodes in the nRE and LFP recording electrodes in the CA1 and mPFC. Mice were trained and tested daily in the delta maze, as previously described, and their performance following general anesthesia was compared to a stable baseline (SWM-T1 vs. SWM-T2). Similar to the first (un-implanted) cohort (Fig. 1d, e), APP/PS1 mice displayed a significant decrease in choice accuracy following anesthesia that mostly recovered after 3 days (Fig. 6a–c and Supplementary Fig. 8a). We then tested whether tDBS-nRE promotes cognitive resilience to a second round of anesthesia in the same mice, using tDBS-nRE in awake, freely behaving mice 5 days prior to the next spatial working memory test (SWM-T3, Fig. 6a). In control (non-DBS) mice, a second session of general anesthesia led to reduced choice accuracy similar to the first anesthesia (Fig. 6d, e and Supplementary Fig. 8b). However, tDBS-nRE prevented the reduction in choice accuracy after the second general anesthesia (SWM-T4, Fig. 6b, c and Supplementary Fig. 8a). Mice performance was unimpaired following anesthesia in the tDBS group in comparison to their performance following anesthesia before tDBS-nRE, unlike control group, for 60 and 90 s delay intervals (Fig. 6c, e). The rate of both CA1 and mPFC IES during anesthesia was significantly lower after tDBS-nRE in the same cohort of mice

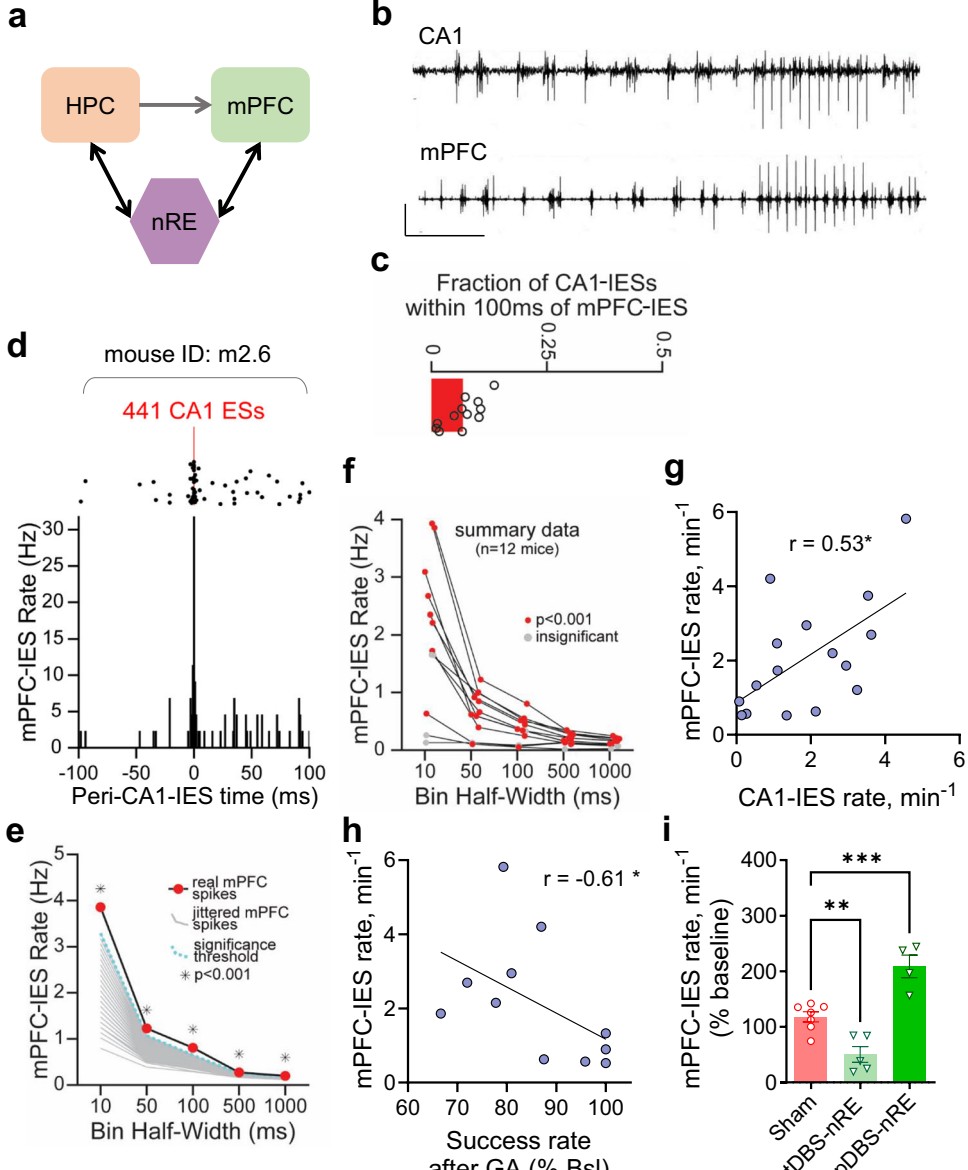

**Fig. 4 | Bi-directional regulation of mPFC hyperexcitability by tonic versus phasic nRE stimulation under anesthesia in fAD model. a** Schematic depiction of the mPFC–nRE–HPC circuit connectivity: The HPC directly connects to the mPFC, with no direct projections from the mPFC to the HPC. The nRE serves as the critical link, forming the HPC–mPFC loop: HPC→mPFC→nRE→HPC. **b** Representative traces show IESs in simultaneous LFP recordings from CA1 stratum radiatum (top) and mPFC (bottom). Scale bars: 0.5 mV, 10 s. **c** Fraction of CA1 IESs within 100 ms of an mPFC-IES in APP/PS1 mice. **d** Cross-correlation between mPFC-IES and CA1-IES times in a single mouse. *Top*, raster plot of mPFC-IESs within 100 ms of CA1-IESs. *Bottom*, peri-CA1-IES time histogram. **e** Calculation of the mPFC-IES rate for the same mouse, focusing on various bin sizes around CA1 spikes. Grey lines represent rates from 1000 jittered mPFC spike trains (see methods). **f** Summary data from each mouse illustrating the mPFC-IES rate in different bin sizes around CA1-IESs. Significant values are marked in red, while insignificant ones are in grey. X-values are jittered for clarity. **g** Correlation between the mPFC-IES rate and the CA1-IES rate in the same mice during the same GA session (Spearman r = 0.53, P = 0.037, n = 16; 13 males, 3 females). **h** Inverse correlation of the mPFC-IES rate with spatial working memory after GA, normalized to baseline success rate, in the 90 seconds delay (Spearman r = −0.61, P = 0.049, n = 11 males). **i** The effect of tDBS and pDBS of the nRE on the mPFC-IES rate during GA (P < 0.0001, One-way ANOVA with Sidak's multiple comparison tests: Sham vs Tonic P = 0.0051, Sham vs Phasic P = 0.0009, n = 7, 5, and 4 for Sham, tDBS, and pDBS, male/female ratio: 5/2, 4/1, 3/1, respectively). Error bars represent SEM; ns non-significant, *P < 0.05, **P < 0.01, ****P < 0.0001. P-values of (**e**, **f**) are in the source file.

(Fig. 6f,g). Physiological parameters of anesthetic depth, such as burst-suppression rate (Fig. 6h), heart rate, and respiratory rate (Supplementary Fig. 8d, e), were similar before and after tDBS. Moreover, tDBS-nRE prevented a reduction in paired-pulse ratio induced by anesthesia, maintaining it at 1.9 ± 0.11 (Fig. 6i–k). These results demonstrate that tonic nRE stimulation rescues anesthesia-induced impairments of synaptic facilitation and spatial working memory in APP/PS1 mice.

## tDBS-nRE prevents age-dependent memory decline

Finally, we asked whether the hyperexcitability exposed by anesthesia is solely responsible for transient memory impairments during the prodromal disease stage or if it also unmasks inherent age-related cognitive deficits at later stages. Specifically, we tested whether the thalamic nRE plays a role in the age-dependent decline of working memory during later disease stages. To address this question, we performed tDBS-nRE in young 4-5-m.o. APP/PS1 mice, before the onset

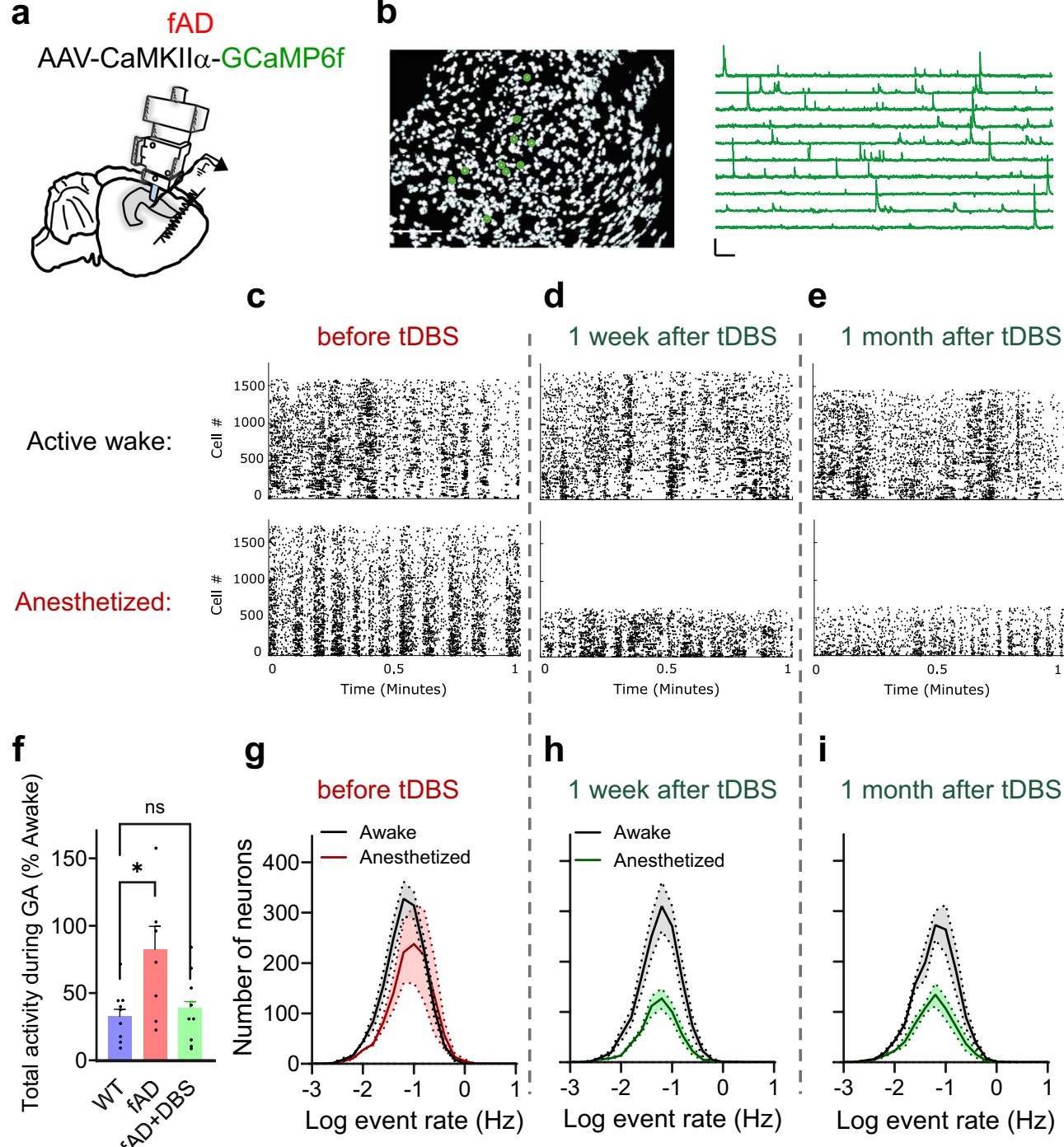

**Fig. 5 | Tonic nRE stimulation restores anesthesia-induced dyshomeostasis of CA1 activity in fAD mice. a** Illustration of head-mounted miniaturized fluorescence microscope in the hippocampus of fAD mice. **b** Representative results of CNMF-E method, as repeated for the 3 sessions * 3 mice. *Left*: Spatial locations of hundreds of cells that were identified using CNMF-E method. Scale bar: 100 μm. *Right*: Relative fluorescence change traces of 10 cells representing Ca²⁺ transients in the CA1 excitatory neurons of fAD mice. Scale bars: 5 min, 10 z-scores (right). **c**–**e** Representative raster plots demonstrating CA1 Ca²⁺ event rates during awake (top) and general anesthesia (bottom) states before (**c**), 1 week (**d**) and 1 month (**e**) after tDBS-nRE. **f** Single-neuron activity (mCaR*Na) during anesthesia, normalized to the activity during awake at the same day, was higher in fAD mice compared to WT before DBS, but similar to WT after DBS. (*P* = 0.0394, Kruskal-Wallis test with Dunn's multiple comparison tests: WT vs. fAD *P* = 0.0347, *n* = 3, WT vs. fAD+DBS *P* = 0.9566, *n* = 3. 1 male, 2 females; dots represent separate recording sessions). **g**–**i** Effect of general anesthesia on average Ca²⁺ event rate distributions of CA1 neuronal populations in fAD mice (5,355 cells, 3 mice;1 male, 2 females) before (**f**), 1 week (**g**) and 1 month (**h**) after tDBS-nRE. Error bands & bars represent SEM; ns non-significant, *P < 0.05.

of working memory dysfunctions, and compared their cognitive performance to a sham control group (Fig. 7a). We implemented the stimulation protocol monthly until the mice reached an older age, characterized by robust SWM deficits. After training (Supplementary

Fig. 9), untreated 8-9-m.o. APP-PS1 mice (-DBS old) exhibited significantly lower success rates in the delta maze at both 60-second and 90-second delays compared to the control group of young APP/PS1 mice (-DBS young, Fig. 7b, c). Notably, tDBS-nRE, initiated at 4-5

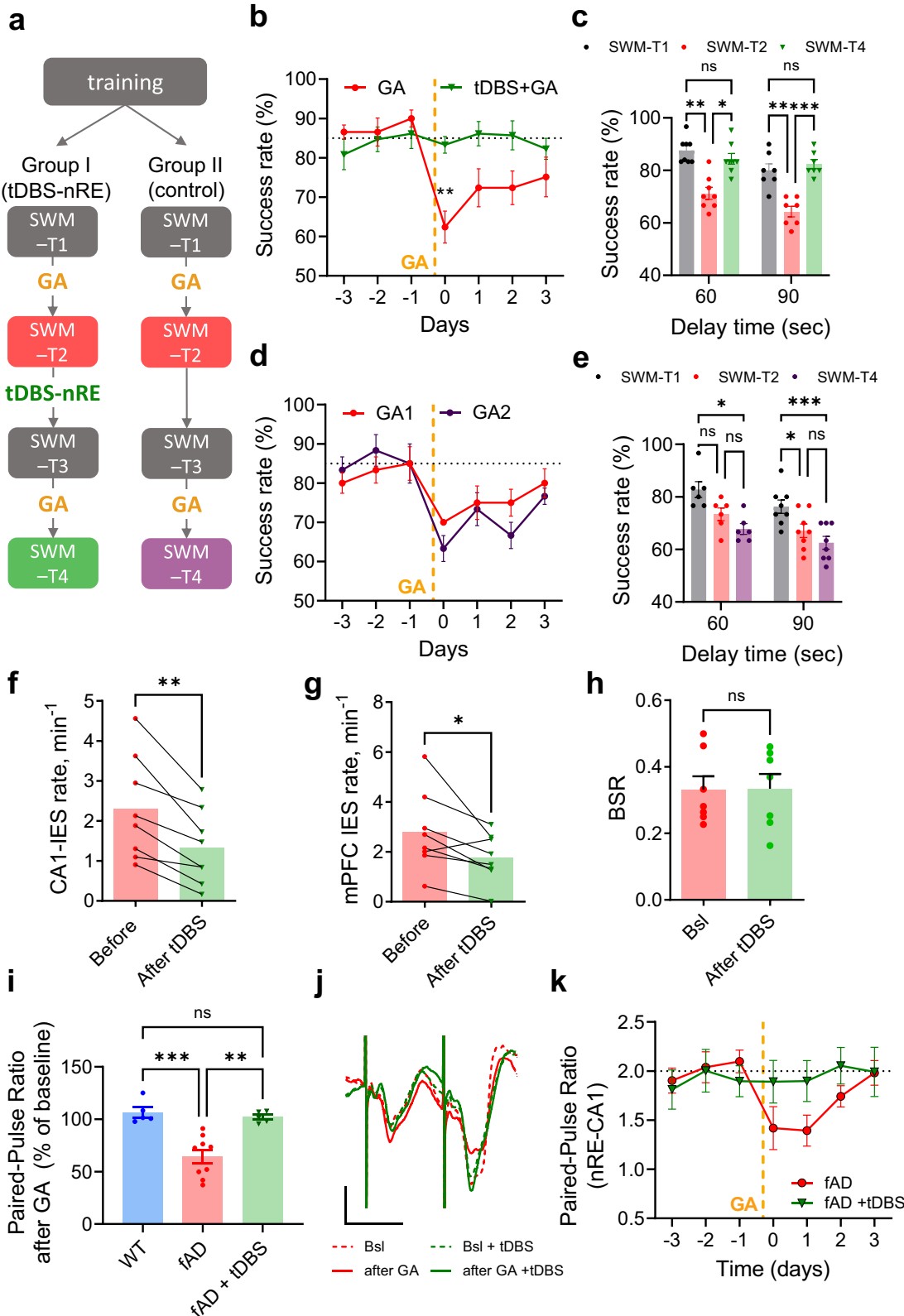

months, prevented the age-dependent decline in delta maze performance at 8-9 months in APP/PS1 mice (Fig. 7b, c). Furthermore, we observed an inverse correlation between the rate of IESs during anesthesia in young APP/PS1 mice and working memory at an older age for both, 60-second and 90-second delays in the same mice (Fig. 7d). These results indicate that tDBS-nRE during the prodromal disease stage can effectively prevent age-dependent memory impairments.

## Discussion

AD is characterized by a prolonged presymptomatic phase that precedes the onset of clinical symptoms[1,2]. Furthermore, ~30% of individuals who show neuropathological features of AD do not develop dementia in their lifespan[3]. These findings point to the existence of cognitive resilience mechanisms that counteract AD progression. Large efforts have been devoted to understanding resilience to AD,

**Fig. 6 | Tonic nRE stimulation rescues anesthesia-induced impairments of working memory and short-term synaptic plasticity in fAD mice.**
**a** Experimental design. SWM-T spatial working memory test; GA general anesthesia. **b** Temporal dynamics of fAD mice's SWM pre- and post-tDBS-nRE for 60 s delay ($n = 8$) ($P = 0.0098$ for day 0 after GA, Sidak's multiple comparisons). **c** GA-induced SWM impairments rescued by tDBS-nRE ($P = 0.0001$, mixed-effect analysis, Sidak's multiple comparisons: 60 s delay: $P = 0.0021, 0.0306, 0.7189$ for baseline vs. 1st GA, GA1 vs. GA2, baseline vs. 2nd GA, respectively ($n = 8$). 90 s delay: $P = 0.0040$, $0.0005, 0.7774$ for the same comparison groups, $n = 7$). **d** Temporal dynamics of fAD mice's SWM after 1st (red) and 2nd (purple) GA session for 60 s delay ($n = 6$). **e** GA induced similar SWM impairments in 1st and 2nd anesthesia sessions ($P = 0.0001$, mixed-effect analysis, Sidak's multiple comparisons. 60 s delay: $P = 0.1104, 0.1547, 0.0114$ for baseline vs. 1st GA, GA1 vs. GA2, and baseline vs. 2nd

GA; $n = 6$; 90 s delay: $P = 0.0114, 0.1051, 0.0004$ for the same comparison groups; $n = 6$). **f–g** Lower IES rates in CA1 (f, $P = 0.0078$, Wilcoxon two-tailed test, $n = 8$) and mPFC (g, $P = 0.0469$, Wilcoxon two-tailed test, $n = 7$) after tDBS-nRE. **h** No change in burst-suppression ratio (BSR) after tDBS-nRE ($n = 7$, $P = 0.9669$, Wilcoxon two-tailed test). **i** Paired-pulse ratio in nRE-CA1 pathway of WT ($n = 5$; 3 males, 2 females), fAD ($n = 9$; 6 males, 3 females), and fAD+tDBS-nRE ($n = 5$; 3 males, 2 females) for 3 days post-GA, normalized to baseline ($P = 0.0001$, one-way ANOVA, Sidak's multiple comparisons: WT vs. fAD $P = 0.0004$, fAD +tDBS vs. fAD $P = 0.0011$, WT vs. fAD +tDBS $P = 0.9522$). **j** Representative fEPSP traces in nRE-CA1 synapses during baseline (dashed line) and 24 hr post-GA, before (red) and after (green) tDBS-nRE. Scale bars: 0.2 mA, 30 ms. **k** Paired-pulse ratio in nRE-CA1 pathway of fAD mice during baseline and 4 days post-GA before (red) and after (green) tDBS ($n = 5$). Error bars represent SEM; ns non-significant, *$P < 0.05$, **$P < 0.01$, ***$P < 0.001$.

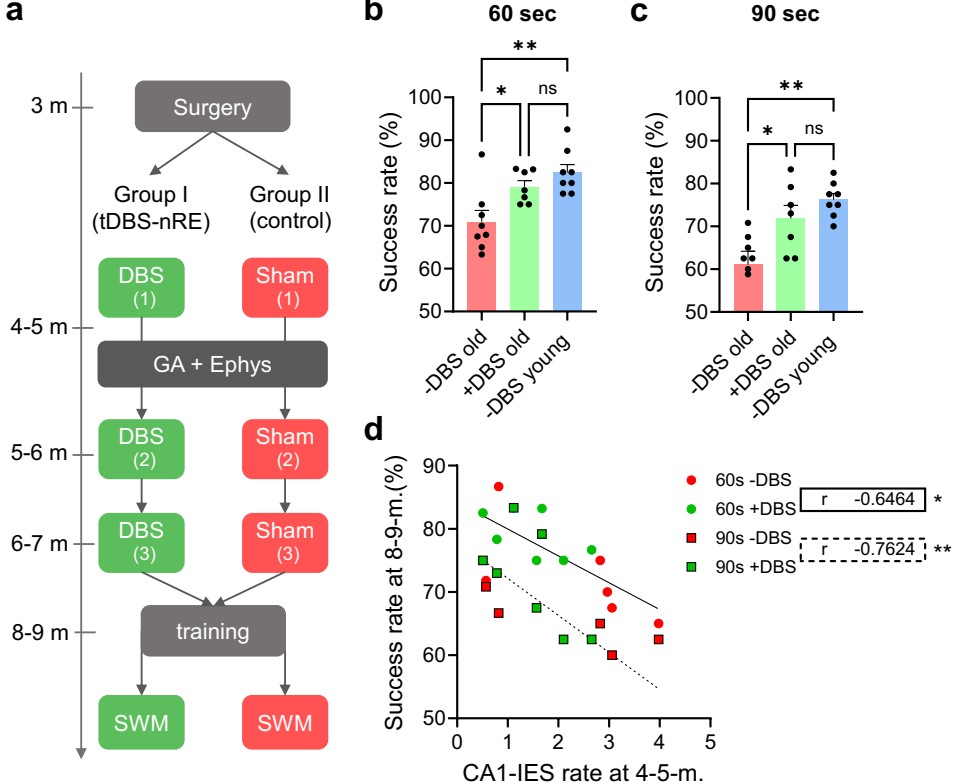

**Fig. 7 | Tonic nRE stimulation started during the prodromal disease stage prevents later impairments of spatial working memory in fAD mice.**
**a** Experimental design: SWM test- spatial working memory test via delta maze. GA: 3 h of 1.5% isoflurane. Ephys – LFP recording and IES analysis at age 4–5 months, after the first DBS. **b, c** Impairments of SWM in old (8–9-months APP/PS1 mice: -DBS old) compared to young (4–5-months. AP/PS1 mice: -DBS young) in the delta maze was prevented by tDBS-nRE (+DBS old). One-way ANOVA with Sidak's multiple

comparison tests: 60 sec delay: -DBS old vs +DBS old: $P = 0.024$, -DBS old vs -DBS young: $P = 0.002$; $n = 7$ per group old (5 males, 2 females) and $n = 8$ young (males). 90 s delay: -DBS old vs +DBS old: $P = 0.018$, -DBS old vs -DBS young: $P = 0.0013$. **d** Frequency of CA1-IES during GA in APP/PS1 mice at young age inversely correlates to spatial working memory performance at the 60 seconds delay (Spearman $r = -0.65$, $P = 0.0198$) and 90 s delay (Spearman $r = -0.76$, $P = 0.0035$) of the same mice at old age. Error bars represent SEM; ns non-significant, *$P < 0.05$, **$P < 0.01$.

focusing on molecular, cellular, and biochemical pathways involved in amyloid and tau pathology clearance[70]. Nevertheless, synaptic and neuronal circuit activity also plays a crucial role in AD-related protein dyshomeostasis[71–74] and cognitive decline[75–77]. In this work, we studied the circuit mechanisms that contribute to cognitive resilience in the face of amyloid pathology in a mouse model of fAD.

Resilience is defined as the ability of a system to maintain homeostasis in the face of a perturbation. This concept encompasses two vital components: (1) The sensitivity to a perturbation, which describes how easily a system can deviate from its stable state, or 'set point'; and (2) Homeostatic capacity for recovery, which enables a precise adaptive response to return the system to its original set point after a perturbation, thus preventing or delaying downstream dysfunctions.

Using general anesthesia as a perturbation, we identified four dormant dysfunctions in otherwise cognitively unimpaired fAD-model mice: (1) CA1 and mPFC hyperexcitability manifested in the form of IESs, (2) dysregulation of CA1 firing rate homeostasis, (3) decreased nRE-CA1 short-term facilitation, and (4) working memory impairments. Understanding these dormant dysfunctions and their resilience to AD-related pathology provides valuable insights into the early stages of AD and potential targets for therapeutic interventions.

Can hippocampal hyperexcitability observed during anesthesia in the prodromal disease phase serve as a predictor of cognitive decline during the ensuing symptomatic disease stage? To address this question, we conducted a longitudinal study where we implemented tonic stimulation of the nRE (tDBS-nRE) starting at the prodromal

stage and continued on a monthly basis until the symptomatic disease stage. Our findings revealed that tDBS-nRE, which successfully prevented IESs and abnormal firing rates during anesthesia, also effectively prevented cognitive decline during the later symptomatic stage. Additionally, we discovered a negative correlation between the rate of IESs during anesthesia in the prodromal stage and working memory success during the later disease stages in fAD mice. This suggests that anesthesia is indeed unmasking IESs and that these IESs contribute to cognitive decline later on. These results lead us to the conclusion that IESs occurring during anesthesia could serve as predictive markers of later cognitive decline. As our present study exclusively involved behavioral experiments in male mice, future research is needed to address the efficacy of tDBS-nRE in female model mice. Furthermore, whether the rate of IESs during anesthesia in presymptomatic AD patients can predict the severity of cognitive decline at later disease stages and whether cortico-hippocampal IESs play a role in POCD are still open questions that will require further investigation. The data also suggest that IESs that occur spontaneously during natural sleep in prodromal stages[26,78–80] are also pathological.

How might nRE synaptic activity contribute to cognitive impairments? The nRE has been implicated in SWM by serving as a conduit of reciprocal communication between the mPFC and CA1[51]. Theoretical studies suggest that stimulus-selective patterns of short-term synaptic facilitation can maintain working memory[56,81] independently of spiking activity[82]. Indeed, SWM impairments following anesthesia were accompanied by a decrease in synaptic facilitation at the nRE-CA1 synapse. However, these functional changes were not associated with alterations in soluble Aβ40 and Aβ42 peptides. The lack of anesthesia-induced hippocampal Aβ changes in this study could be attributed to the fact that anesthesia did not cause an increase in the CA1 mean firing rate in APP/PS1 mice, while Aβ release/production is known to positively correlate with firing rate and synaptic vesicle release[71–73]. Molecular mechanisms underlying maladaptive plasticity of thalamo-hippocampal synapses in the mouse model of amyloidosis remain unknown. Future research is required to link the anesthesia-induced transcriptional, synaptic, and cognitive dysfunctions. Notably, the snRNAseq analysis of excitatory CA1 neurons revealed that many of the top differentially expressed genes are related to synaptic function and neurodevelopmental disorders. As epileptiform activity is a common feature in various neurodevelopmental[83,84] and neurodegenerative[78] disorders, IESs may induce some shared signaling pathways underlying cognitive dysfunctions in these conditions.

We have identified the nRE of the midline thalamus as a key node in a neural circuit responsible for regulating resilience to AD. Various manipulations of nRE activity - pharmacological inhibition of nRE spiking, chemogenetic inactivation of nRE-CA1 synapses, and tDBS-nRE - all suppressed CA1 hyperexcitability under anesthesia in fAD mice. Conversely, pDBS-nRE exacerbated CA1 hyperexcitability. The suppression of hyperexcitability by tDBS-nRE resulted in a reduction of CA1 and mPFC epileptiform discharges, along with a restoration of state-dependent regulation of CA1 firing rates. Most importantly, this tDBS-nRE intervention effectively prevented both the synaptic and cognitive aftermath of anesthesia, as well as age-dependent memory decline. Remarkably, these positive outcomes were observed even when tDBS-nRE was applied to awake, freely behaving mice. Interestingly, thalamic TRN, which serves as the main inhibitory input to the nRE, has been implicated in AD-related sleep disturbances[85]. Thus, the nRE might play a crucial role in dysregulation of cortico-hippocampal activity induced by diverse low-arousal brain states, including sleep, in AD.

In summary, this study demonstrates that a short and feasible intervention targeting the nRE during the prodromal disease stage can restore synaptic, circuit and cognitive resilience to amyloid pathology in a mouse model of fAD. While the research was conducted in mice, the therapeutic potential of this approach in humans could be enormous. For instance, DBS of the subthalamic nucleus has already shown success in improving motor function in advanced Parkinson's Disease[86]. However, its effectiveness in addressing cognitive dysfunctions remains uncertain and controversial[87]. In contrast to the high-frequency (130-185 Hz) continuous stimulation commonly employed for Parkinson's disease treatment[88], our work demonstrates that brief, tonic stimulation at an intermedium frequency (25 Hz) provides protection against neuronal, synaptic and cognitive impairments in the fAD model. Notably, high-frequency phasic stimulation (100 Hz in-burst frequency) exacerbates epileptiform activity in this context. Optimizing the DBS protocol will be crucial to evaluate the potential of tDBS-nRE in confering cognitive resilience to patients with POCD and AD. It remains to be seen whether rescuing activity homeostasis in the nRE-CA1-mPFC circuitry by tonic DBS of the midline thalamic nRE might delay the transition from dormant to clinical AD.

## Methods

### Mice

All animal experiments detailed herein were approved by the Institutional Animal Care and Use Committee (IACUC) of Tel Aviv University (protocol numbers: TAU-MD-IL-2204-141-5 and 01-17-099). Experiments were performed on 4-6 month old male and female APP/PS1 (APP$_{Swe}$/PS1ΔE9) hemizygotes (Stock No. 005864, The Jackson Laboratory) and their wild-type littermates. As no disparities in electrophysiological and physiological measurements were discerned between the sexes (Supplementary Fig. 10), both male and female mice were used for electrophysiological and Ca$^{2+}$ imaging measurements, and for behavioral experiments in Fig. 7. Males only were used for behavioral experiments in Fig. 1c–e and Fig. 6. All mice were on a C57BL/6J-congenic background. All animals were kept in a normal light/dark cycle (12 h/12 h, lights on at 7AM). All animals except for mice undergoing delta maze experiments had free access to food and water. Transgenic and wild-type mice were housed together for behavioral and histological experiments. Mice for electrophysiology and Ca$^{2+}$ imaging that didn't participate in behavioral experiments were singly housed after the surgery.

### General surgical procedures

In all the surgical procedures, mice were anaesthetized with 5% iso-flurane (Sigma Aldrich, CAS: 26675-46-7) by volume for induction, injected i.p. with ketamine/xylazine (60 mg/kg ketamine, Sigma Aldrich, CAS: 1867-66-9 and 5 mg/kg xylazine, Sigma Aldrich, CAS: 23076-35-9), head fixed to a stereotaxic apparatus (David Kopf instruments) and then maintained anesthetized by continuous iso-flurane (~1.25%) inhalation throughout the surgical procedure. Eye ointment was used to protect the mice eyes (Duratears, Vetmarket) and body temperature was recorded and maintained by a heating pad (FHC, DC temperature controller) at 34 °C throughout the surgery. At the beginning of each surgical procedure the mice were injected subcutaneously with Carprofen (5 mg/kg, Sigma Aldrich, CAS:53716-49-7) to reduce inflammation and pain. At the end of each surgical procedure the mice were injected subcutaneously with Buprenorphine (0.05 mg/kg, Sigma Aldrich, CAS:52485-79-7) to reduce pain. The mice were then allowed to recover in their home cage for at least 1–2 weeks before the subsequent surgical procedure or experiment began.

### LFP / fEPSP recordings

Small diameter holes were drilled in the skull at the position of the recording and stimulating electrodes. The recording electrode (bipolar stainless steel; 0.1397 mm diameter with coating, 0.0762 mm bare diameter) was slowly lowered through the cortex into the CA1 stratum radiatum (2.06 mm posterior to bregma; 1.5 mm mediolateral, ML; 1.5 mm dorsoventral from bregma, DV) or the mPFC (1.75 mm anterior to bregma; 0.5 ML; 1.6 mm DV), and the stimulating electrode (bipolar stainless steel; 0.1397 mm diameter with coating, 0.0762 mm bare

diameter) was slowly lowered through the cortex into the nRE (0.82 mm posterior to bregma; 1.1 mm ML; 4.55 mm DV with an angle of 15 degree) or the SC (2.54 mm posterior to bregma; 2.75 mm ML; 2.2 mm DV). Ground electrode was screwed to the skull above the cerebellum. Test stimuli of 250 or 50 μA were delivered to the nRE and SC (respectively) at 0.06 Hz while the positions of the stimulation and recording electrodes were slightly adjusted to optimize the signal. The electrodes were firmly fixed to the skull with dental cement (C&B Metabond, Parkell) and dental acrylic. A metal bar was added to the posterior aspect of the construct to head fix the animal when needed. Operated mice recovered in their home cage for at least a week following electrode implantation. For evoked fEPSP recordings in awake mice (Figs. 3 and 6i–k), mice were habituated for a few days to the experimental device that was composed of a running wheel and a metal bar, which the mice were head fixed to throughout the experiment with a screw.

Extracellular field potentials were amplified ×100 using Model 1700 by A-M Systems amplifier, band-pass filtered between 0.1 Hz and 5 KHz, and digitized by Digidata 1440 A at 56 kHz sampling rate (Molecular Devices). Data was analyzed using Clampfit 10.7 (Molecular devices) for fEPSPs or custom MATLAB functions (MathWorks).

## Surgical procedure for $Ca^{2+}$ imaging

The surgical procedures were previously described[67]. First, 500 nL of the viral vector AAV5-CaMKII-GCaMP6f (prepared by University of North Carolina Vector Core) was injected into the CA1 pyramidal layer at the following coordinates: −2.18 mm AP, -1.5 mm ML, and -1.3 mm DV to bregma. The skin was sutured and disinfected using Betadine solution. Two weeks after virus injection, a glass guide tube was implanted directly above CA1. For this, a trephine drill was used to remove a circular part of the skull located posterior-lateral to the viral injection site, and the dura, cortex and the hippocampal commissures above the CA1 were removed by suction with a 29 gauge blunt needle while constantly washing the exposed tissue with sterile PBSx1 (Sigma Aldrich, CAS:7758-11-4). A glass guide tube was then implanted above the CA1 stratum pyramidale. A recording electrode (bipolar stainless steel; 0.1397 mm diameter with coating, 0.0762 mm bare diameter) was slowly lowered through a hole that was drilled adjacent to the guide tube into CA1 stratum radiatum (1.5 mm DV). The stimulating electrode (bipolar stainless steel; 0.1397 mm diameter with coating, 0.0762 mm bare diameter) was slowly lowered through the cortex into the nRE (0.82 mm posterior to bregma; 1.1 mm ML; 4.55 mm DV with angle of 15 degree). Ground electrode was screwed to the skull above the cerebellum. Test stimuli of 250 μA were delivered to the nRE at 0.06 Hz while the positions of the stimulation and recording electrodes were slightly adjusted to optimize the signal. The electrodes were firmly fixed and the remaining exposed area of the skull was covered with dental cement (C&B Metabond, Parkell) and dental acrylic (Henry Schein). A PLA+ bar was added to the posterior aspect of the construct to head fix the animal when needed. Two stainless-steel wires were inserted to either side of neck muscles, and referenced bipolarly, to measure EMG activity. 5 kHz noise in the LFP/EMG recordings generated by the electronic focus of the nVista3 microscope was filtered out by a band-pass filter (1-300 Hz).

## $Ca^{2+}$ imaging in behaving mice

For time-lapse imaging in behaving mice we used an integrated miniature fluorescence microscope (nVista 3.0, Inscopix) as previously described[26]. At least two weeks after the glass guide tube implantation, we inserted a microendoscope consisting of a metal guide cannula (-3.1 mm length, 1.8 mm outer diameter) and a single gradient refractive index (GRIN) lens (4.0 mm length, 1.0 mm diameter) into the implanted glass tube and examined $Ca^{2+}$ indicator expression in the operated mice (Inscopix data acquisition software, Inscopix). We then affixed the microendoscope within the glass guide tube using

ultraviolet-curing adhesive (Flow-It A3, Pentron). Next, we attached the miniature microscope's magnetic base plate to the dental acrylic surface with the ultraviolet-curing adhesive. At least a day later, the mice were habituated wearing the miniature microscope in their home cage for 30 minutes for 4–5 days. To record mouse behavior in the home cage, we used an overhead monochrome camera (GigE Vision, Basler AG), which we synchronized with the miniature microscope. $Ca^{2+}$ imaging was performed at 10 Hz. Imaging sessions consisted of 18-min-long trials.

## $Ca^{2+}$ imaging and LFP recording in anesthetized mice

For $Ca^{2+}$ imaging and LFP recording sessions during anesthesia, mice were placed in an induction chamber connected to an isoflurane vaporizer (Isotec 5), the gas flow rate was turned on to 0.8 LPM and the vaporizer was set at 5% isoflurane. When the mouse's breathing slowed down and became rhythmic, it was moved to a physiological monitoring system (75-1500 Harvard Apparatus) and the isoflurane level was reduced gradually and slowly to 1.5%. Respiration and heart rate were monitored and analyzed by the physiological monitoring system. To achieve a steady and consistent anesthetic depth, the isoflurane level was slightly adjusted between 1.3%–1.7% when needed in order to maintain a stable LFP activity pattern of 0.2–0.5 BSR and respiratory rate of 45–90 RPM throughout the duration of the anesthesia. Temperature was also maintained by the device at 37 °C throughout the procedure.

LFP activity in the CA1 stratum radiatum was recorded throughout the duration of anesthesia and $Ca^{2+}$ measurements were taken once a stable LFP pattern was observed, -120 min following anesthesia induction. Experiments were performed under 1.3–1.7% isoflurane, combined with oxygen (100%).

Acute local delivery of either TTX (Alamone labs, CAS:18660-81-6), VEH (DMSO, Sigma Aldrich, CAS:67-68-5) or CNQX (Tocris Bioscience, CAS:479347-85-8) and D-AP5 (Tocris Bioscience, CAS:79055-68-8) during the anesthesia sessions was done using a microliter syringe (Hamilton #65460-05) terminating with a 33 AWG blunt end needle (Hamilton #7762-06), which was slowly lowered through the guide cannula. The syringe was controlled by a pump (World Precision Instruments #UMP3 and #MICRO2T) to deliver the drugs at a rate of 70 nL/min. After injection, the syringe was kept in place for an additional 10 min to prevent back propagation of the drug. After each injection, a small volume (100–500 nL) of the drug was released outside of the brain to ensure the needle was not clogged.

Acute i.p. injections of CNO (Sigma Aldrich, CAS:34233-69-7) or saline (Baxter, CAS:7647-14-5) were done using 0.5 ml sterile disposable syringes with 31 G insulin needle (insumed by PIC).

## $Ca^{2+}$ imaging data analysis

We pre-processed the raw imaging data using commercial software (Inscopix data processing software, Inscopix). Raw data was spatially down-sampled by a factor of 2, cropped to 1200 by 840 μm rectangle, motion corrected, and exported as a TIFF file. Single neurons and their calcium signals for each data set were extracted using the constrained non-negative matrix factorization algorithm for endoscopic recordings[89] (CNMF-E). The analysis parameters were set as described earlier:[26] maximal diameter of neurons in the imaging plane was set to 13 pixels (gSiz), and the width of the gaussian kernel, which can approximate the average neuron shape, was set to 3 pixels (gSig). Neurons with spatial overlap ratio greater than 0.85, temporal correlation of calcium traces greater than 0.85 and centroid distance less than 2 pixels (dmin) were merged. Region of interest (ROIs) that had a minimum peak-to-noise ratio for a seeding pixel of 8 (min_pnr) and a minimum spike size of 5 (min, the actual threshold is = |smin*noise level|) were extracted. These parameters were used for all data sets analyzed in the paper. Due to one-photon background fluorescence fluctuations, motion artifacts, and segmented dendrites,

some of the ROIs detected by CNMF-E cannot be considered as true cells, but false positive detections. Therefore, we have inspected the registered ROIs spatial footprints and excluded them based on the following exclusion criteria: minimum peak-to-noise ratio of 8, 300 > ROI size >30 pixels, and minimum circularity estimate of 0.5 (1 equals a perfect circle). Filtered data sets were further manually inspected and verified by the experimenter (~5-10% cells were excluded per data set). CNMF-E "S" output was used as the inferred spiking activity (event rate) obtained from the denoised ("C") scaled version of DF ("C_raw"). "Event rate" of a cell was defined as it summed inferred spiking activity (obtained from the "S" output) throughout the recording session, divided by the recording time. "Total activity" of a session was defined as its cells' median inferred spiking activity value, multiplied by the number of participating cells in the session. Single-cell and population analysis were performed with custom MATLAB functions.

### Anesthesia depth analysis

For detecting burst-suppression (deep anesthesia), raw LFP recordings were filtered between 1 and 1000 Hz. The absolute value of the signal was averaged and divided to bins of 700 ms, that showed a bimodal distribution and thus were used to classify the bins as periods of bursts or suppression. The negative peak of the bins' histogram was used as a threshold in order to sort the bins to burst or suppression. Epochs of bursts were merged if the inter-burst interval was less than 2 s and were accepted as bursts only if their duration was greater than 2 s. Burst-suppression ratio (BSR) was calculated in 1 min bins as the fraction of time in suppression. Overall, this algorithm demonstrated >92% accuracy when compared to the manual scoring of two data sets from different genotypes.

Respiration and heart rate were monitored and analyzed by a physiological monitoring system (75-1500 Harvard Apparatus).

### c-Fos staining

30–60 min post-GA, mice were transcardially perfused with cold PBS followed by 4% paraformaldehyde (PFA) in PBS. The brains were extracted, postfixed overnight in 4% PFA at 4 °C, then rinsed 3 times with 0.1xPBS and kept in PBS. Brains were sectioned to a thickness of 50 μm using Leica microtome, then washed 3 times for 5 minutes in 1xPBS, then in PBST (0.1% TritonX-100). Free-floating sections were washed in PBS, incubated for 1 h in blocking solution (1% BSA and 0.3% Triton X-100 in PBS) and incubated with the primary c-Fos antibody (rabbit, Synaptic Systems, Cat.No. 226 003; 1:10,000) for five nights at 4 °C. Sections were then washed with PBS and incubated for 2 h at room temperature with secondary antibody (goat anti-rabbit, DyLight 488, Jackson Laboratories; 1:600) in 1% BSA in PBS. Finally, sections were washed in PBS, mounted on slides with DAPI mounting medium (Sigma, F6057), and sealed. Imaging was performed by Leica Aperio VERSA8 slide scanner.

### Detection of IESs in fAD models

High-voltage IESs were detected by setting a threshold of 10 $z$-scores above and below the mean voltage during the suppression epochs of the recording and accepted only if their peak-to-peak value within 30 ms was greater than 10 $z$-scores. A dead-time of 50 ms was set to ensure the same spike was not counted twice. Subsequently, all spikes were manually inspected and approved.

A nonparametric significance test based on jittering was used to validate the significance of cross-correlations of mPFC & CA1 IES, as described previously[90,91]. Synchrony was defined as the mPFC IES rate within different bin sizes (10, 50, 100, 500, 1000 ms bin half-widths) centered around each CA1 IES. For each bin size, we created 1000 surrogate jittered mPFC spike trains where each mPFC IES was jittered randomly between +/− 2*half-width. Synchrony at each bin was then calculated for each jittered spike train against the real CA1 spike train. Real CA1-mPFC synchrony values in the 99.999th percentile of CA1-jittered synchrony values were deemed statistically significant ($P = 0.001$).

### Spatial working memory test

Spatial working memory was examined using a continuous variation of the T-maze called Delta-maze[92]. In sum, mice were required to traverse the central arm of a delta-shaped maze and alternate between left and right turns during subsequent trials (Supplementary Fig. 1e). Correct trials were reinforced with a drop of condensed milk at the edge of the selected sidearm. After entering one of the side arms, retracing was prevented by hinged doors such that mice could initialize a new trial only by returning to the base of the central arm via a connecting arm. At the base, mice were confined for a specified delay before the door to the central arm was opened and a new trial commenced. The first trial of each session contained a reward in both side arms of the maze. Prior to the onset of training, mice were habituated to the apparatus for 15 m a day for 5 consecutive days. During training, mice performed 10 trials of the task with a 10 s delay. During testing, the delay varied between days from 10 to 180 s. To facilitate learning and ensure consistent motivation when performing the task, the food intake of mice was restricted such that their weight was kept at 85–90% relative to ad libitum feeding. The success rate was defined as the percentage of correct alternations relative to the number of trials.

### snRNA-seq library preparation and pre-processing

All the mice were rapidly decapitated, then brains were quickly removed, and tissue dissection was performed immediately. The tissue was frozen after dissociation and kept at −80 °C until further processing. Tissue was processed in batches of four with samples from each experimental group of male mice (WT-awake, WT-GA, fAD-Awake, fAD-GA). WT-GA and fAD-GA groups were decapitated 3 h after anesthesia induction. Working on ice throughout the nuclei isolation process, the frozen hippocampus tissue was processed as previously reported[93] with the following changes: NP40 Lysis buffer contained 10 mM Tris HCL pH 7.5, 10 mM NaCl, 3 mM MgCl2, 0.1% NP-40, 40 U/mL of RNAse inhibitor (NEB M0314L), and the tissue was gently dounce homogenized 10 times with pestle A followed by 10 times with pestle B. The snRNA libraries were prepared using the 10× Chromium controller instrument and Single Cell 3′ Reagent Kits v3, following the manufacturer's protocol, and sequenced using NextSeq 2000. De-multiplexing, alignment to the mm10-2020-A transcriptome, and gene quantification were performed using the Cellranger toolkit (v.5.0.0, chemistry V3, 10X Genomics, 3′ chemistry, and --include-introns flag). Empty droplet removal and ambient RNA correction were done using CellBender[94]. Nuclei with fewer than 200 detected genes were filtered out.

### snRNAseq data analysis

SnRNA-seq data was scaled and clustered by Seurat package v.4.1.0[95], following our analysis pipeline previously described[93,96]. Briefly, we performed log-normalization of genes within each sample (NormalizeData function), selection of variable features by variance-stabilizing transformation (FindVariableFeatures function), and scaling genes across samples (ScaleData function). Doublet cells were identified by DoubletFinder (v3)[97], filtering out cells and small clusters of cells with high doublet scores. Batch correction by Harmony algorithm (package v.0.1.0)[98]. Dimension reduction by principal component analysis (PCA, RunPCA function), and top PCs were used for embedding in 2D by UMAP and for clustering by Louvain community detection algorithm on the k-NN neighbor graph (FindNeighbors and FindClusters function). Identification of cell types was done using a logistic regression classifier (linear_model.LogisticRegression from Python's sklearn package) trained on published mouse hippocampus dataset[99] (Trained on 80%, tested on 20%). Clusters were annotated to cell types

according to the classifier's output and validated using marker genes, specifically identifying the CA1 neurons. Differential expression test on CA1 neurons was done using the MAST[100] algorithm corrected for batch (FindMarkers function with test.use = "MAST", latent.vars=Batch, assay=RNA), requiring differential genes to be expressed in at least 10% of nuclei in the given cluster and with at least 0.25-fold change. The differential expression signatures were tested for enriched pathways using a hypergeometric test, and corrected for multiple hypotheses by FDR, comparing against the pathways annotations from KEGG (using clusterProfiler method enrichKEGG, organism=mmu), Reactome (using ReactomePA method enrichPathway organism=mouse) and GO (using clusterProfiler method enrichGO, OrgDb=org.Mm.eg.db, ont=BP, pAdjustMethod=BH), compared to the background set of all expressed genes within our dataset. Enriched pathways were clustered using the simplifyEnrichment package (kappa distance with binary_cut) with the clustering cutoff set by manual inspection.

### Statistical analysis

Error bars shown in the figures represent SEM. All the experiments were repeated at least in three different animals. Statistical significance was assessed by unpaired or paired Student's *t*-tests, Mann–Whitney *U*-tests, one-way analysis of variance (ANOVA), or two-way ANOVA, where appropriate (multiple comparison tests are specified in figure legends). Normality was assessed using the Shapiro–Wilk test. For non-normal distributions, differences between groups were tested with the Wilcoxon signed-rank test for paired data and Mann–Whitney test for unpaired data. Correlation was assessed using the Spearman test. A comparison of distributions was performed by Kolmogorov–Smirnov test. Statistical analysis was performed using Prism 9.0 GraphPad. The statistical test, *P* value, and the number of cells / mice that went into the calculation are reported in figure legends. Significance was declared at $P < 0.05$ and all tests were two-sided.

### Reporting summary

Further information on research design is available in the Nature Portfolio Reporting Summary linked to this article.

## Data availability

The authors declare that the main data supporting the findings of this study are available within the article and its Supplementary Information files. The raw and processed sequencing data generated in this study is publicly available in the Gene Expression Omnibus database under accession code number GSE245201. Extra data are available from the corresponding author upon request. Source data are provided with this paper.

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

## Acknowledgements

We thank Leore Heim for comments on the manuscript and all members of the Slutsky Lab for fruitful discussions. This work was supported by research grants from the European Research Council (724866 and 101097788 to I.S.), the MOST-IL-China research (3-15687 to I.S. and N.H.), the Israel Science Foundation (1663/18 to I.S.), The Deutsche Forschungsgemeinschaft (440813539 and 448865644 to I.S.), BIRAX Regenerative Medicine Initiative (46BX18TKIS to I.S.), Rosetrees Trust (to I.S.). N.H. is a Goren Khazzam senior lecturer in neuroscience, supported by the Myers Foundation (N.H.), the Israel Science Foundation (1709/19) and the European Research Council (853409) research grants. I.S. is grateful to Sheila and Denis Cohen Charitable Trust of the UK for their support. S.S. is a recipient of the PhD fellowship of excellence from Israel Council for Higher Education. T.L. is a recipient of the Post-Doctoral Fellowships from Zuckerman and Fulbright foundations. This work was performed in partial fulfillment of the requirements for a Ph.D. degree by Shiri Shoob at the Sackler Faculty of Medicine, Tel Aviv University, Israel.

## Author contributions

I.S. and S.S. designed the project; S.S. performed all the electrophysiological experiments and analysis, with help of H.B., T.L. and D.Z.; N.B. performed most of the behavioral experiments; O.S. performed calcium imaging experiments, with help of S.S; O.G. and N.H. performed and analyzed the snRNAseq experiments; I.Sh. performed ELISA experiments; G.B. performed immunocytochemistry; I.S. supervised the project; I.S. and S.S. wrote the manuscript with inputs from all the authors.

## Competing interests

The authors declare no competing interests.
