## [Peer Review File · Nature Communications]

REVIEWER COMMENTS

Reviewer #1 (Remarks to the Author):

The studies by Shoob et al. revealed a neural circuit mechanism by which anesthesia dampens resilience to cognitive decline in Alzheimer's disease (AD). The works are significant by demonstrating the circuit mechanisms that underlie the resilience of AD cognitive impairment. More importantly, the studies used anesthesia as a clinically relevant tool to provide clinically relevant findings. The conclusion was supported by the data presented. The methodology was sound, and the data analysis was fine. However, there are some concerns.

1. The studies can be significantly improved by including biochemistry studies. Are there any changes of neurotransmission, e.g., NMDA, Tau, Abeta, or neuroinflammation, that are responsible for the changes in the neural circuit between nucleus reunions and hippocampus CA1 following anesthesia?
2. The pathogenesis of postoperative cognitive dysfunction is more complicated than anesthesia alone. More clinical studies suggest that anesthesia alone may not be associated with postoperative cognitive dysfunction. Thus, the authors need to tone down the conclusion that the circuit between nRE and CA1 serves as the pathogenesis of postoperative dysfunction.
3. Page 8, line 208: What is the purpose of the second session of general anesthesia?
4. Page 9, line 236, the statement may not be relevant to the scope of the current studies. Please see comment two as well.
5. Page 14, line 451-453: Why only male mice were used in behavioral studies, but both male and female mice were used in EP and calcium imaging studies?

Reviewer #2 (Remarks to the Author):

Review of Nature Communications: Shoob et a.

This elegant study of a neuronal system involved in AD pathophysiology and cognitive failures is interesting for a number of reasons. First of all, it is a system study of an AD model. While we often pay lip service to the idea that AD is a failure of neuronal systems, there are few studies of interactions between brain areas involved in the pathophysiology and cognitive symptoms of the disease that also study the mechanisms of those failures at the small neuronal assembly level. For example, studies relating cognitive decline to plaque load and tangles reveal that the relationship to abnormal protein accumulation and aggregation alone cannot completely account for the symptoms of the disease. Clearly, something more complex is involved, and interactions among neurons and brain area activity are a logical place to look (as first demonstrated many years ago by the Malinow lab in vitro). This is a point which should be emphasized in the discussion. Here we have an in vivo comparison of brain activity in the thalamic nRE-hippocampus-PFC system between a well-characterized APP mouse model and controls.

It is clear that AD pathophysiology is a complex matter: there are many papers on hyperexcitability of cortical neurons in APP model mice, but they fail to demonstrate convincing mechanisms, and intracellular recordings in vivo demonstrate that the hyperexcitability is limited to very restricted conditions depending on the state of the neuron. In this study, the authors demonstrate that hyperexcitability of hippocampal neurons is driven by inputs from the nRE and emerges during anesthesia, revealing that AD-model related hyperexcitability is not a simple matter of reduction of cortical inhibition, but is dependent on complex interactions among neurons in different areas. This paper is a timely study of disruption of system homeostasis as a mechanism involved in AD-model pathophysiology and cognitive decline.

The experiments themselves are well designed. I think that some additional analysis might be appropriate. For example, I would like to see the frequency spectra of the CA1 and PFC activity during the different sections of the experiment. This might shed some light on the specific mechanisms involved in the differences in levels of hyperactivity as well as providing information about levels of anesthesia. The Extended data would be a good place for such a figure,

That being said, there are a few issues with the manuscript. One is that I found it difficult to read. It would be helpful to have small subtitles on the figure. For example, The title of Figure 2, is "...CA1 and mPFC hyperexcitability...", but I do not see any data relating to the PFC in that figure. If I am missing something, so will the other readers. In any case, such insertions would greatly improve the readability of the study.

In any case, given that one of the major points is the interaction of the PFC with the other two brain areas, it would be nice to see some recording of those areas performed simultaneously with those made in CA1, as shown in Figure 3 of the extended data. (I would say that that figure should be in the paper itself, as it is extremely important to the study.)

For me, the major issue with the manuscript is the fact that the only anesthetic used is isoflurane. Isoflurane has been shown to induce amyloid accumulation and apoptosis (Xie et al, J Neurosci. 2007 27:1247-1254), as well as affecting physiology of the cortical network in different ways than do other anesthetics. (See differences in EEG under isoflurone and halothane, for example). At the very least, the authors should address that issue in the discussion. However, in my opinion, if they could perform experiments similar to those in described in Figure 3 in the extended results, as well as figures 2 and 3 in the paper. under a different inhalant (or other) anesthetic, the study would be far more powerful. (I do realize that this may be difficult to do in a timely manner, as it may involve an addendum to animal protocols.) If this is not possible, it does not reduce the value of the results (as long as a qualifier is inserted to the effect that it may not be generalizable to all anesthetics; the study is important for the reasons stated above. However, if the effects of a second anesthetic are measured and compared, it would greatly increase the validity of the study.

Reviewer #3 (Remarks to the Author):

The manuscript by Shoob and colleagues describe a set of experiments linking epileptiform spikes recorded in the hippocampus under anesthesia in prodromal AD-model mice to aberrant regulation of input from the nucleus reunions of the thalamus. The authors employ a multi-disciplinary approach to convincingly demonstrate that dysregulation in this circuit, unmasked by anesthesia, contributes to deficits in spatial working memory observed post-anesthesia in AD model, but not WT, mice. Overall the experiments are well thought out and executed, and I have only minor technical comments. My chief criticism for the work presented stems from the difficulty understanding the "big-picture" question that is actually addressed by the studies.

The logic presented in the abstract and introduction suggests that the authors believe the results applicable to resilience against cognitive decline in AD. What is being assessed, however, is a neuro-behavioral phenotype observed prior to the onset of cognitive impairments known to exist in this particular humanized APP mouse line. Independent of known problems with the amyloid-beta hypothesis of AD, properly speaking this would make it a study of predictors of cognitive decline in prodromal disease phases. A question that flows naturally from this would be whether or not anesthesia

is indeed unmasking a mechanism of cognitive impairment that is relevant to later-stage disease? It would seem important then to demonstrate that there is higher epileptiform activity in older, cognitively impaired mice. Would we then suspect that these impairments would also be reversed by the tonic stimulation protocol they describe? It also seems important to identify, mechanistically, what exactly the anesthesia unveiling. What underlies the ability of isoflurane to uniquely tax this nRE-CA1 circuit? Is the effect specific to ISO? What are potential neuropharmacological mechanisms?

With respect to this as a model of the vulnerability of prodromal AD patients to the negative side effects of anesthesia, it would seem important to identify the long term implications, e.g. do mice exposed to anesthesia in the prodromal phase have greater impairment later? Are you tapping into a circuit that later contributes to cognitive impairments, or this specific to the prodromal phase?

It also seems important to determine if these same circuit and behavioral effects are specific to the AD context, or if the same is observed in aged mice, where cognition declines and vulnerability to the negative effects of anesthesia increases. If ES spikes are the mechanism underlying SWM deficits, can you induce them in a WT mouse and create SWM deficits?

In all, the manuscript as presented struggles from a bit of an identity crisis, which ultimately limits its overall impact.

Minor points:

Ca imaging: What was actually quantified here? How many cells per animal? Most of this appears to be ANOVAs run on all cells from all mice. Begs the question of how consistent this is across mice. A more appropriate analysis would be something like a linear mixed effect model with mouse and imaging session as random effects. For the accompanying figure, this data really needs to be shown side by side with the data from WT mice.

For the tonic stimulation: why use this as opposed to DREADD-based inhibition. The latter is a little easier to understand mechanistically, could be useful for contextualizing what the tonic DBS is doing.

Relationship between mPFC ES and SWM? Detail the relationship between mPFC ES and SWM, how does it overlap with hippocampus ES relationship to behavior, where does it differ?

We thank the reviewers and the editor for their time and effort and their constructive feedback on our manuscript (NCOMMS-22-41987-T) “Deep Brain Stimulation of Thalamic Nucleus Reunions Promotes Neuronal and Cognitive Resilience in Alzheimer’s Model”. We were very pleased to learn that all three reviewers evaluate our study as elegant, timely, solid and clinically-relevant. The reviewers suggested us (1) to test if the circuit mechanism that we discovered is relevant to age-dependent cognitive decline at later disease stages; (2) to test if pDBS of nRE also suppresses epileptiform activity exposed by different general anesthetics; (3) to perform additional analyses of epileptiform activity in mPFC and test its impact on working memory impairments; (4) to describe molecular phenotypes, in addition to electrophysiological and behavioral phenotypes, induced by general anesthesia in APP/PS1 mice. Based on these comments, we performed the following additional experiments and analyses. We believe that these additional data and explanations convincingly address the reviewers’ concerns and strengthen the impact of our study.

We first provide a summary of the major experimental additions we made for the revised manuscript, followed by a point-by-point reply to address individual reviewer comments.

- (1) We conducted a new, longitudinal study to test the impact of interictal epileptiform spikes (IESs) exposed by anesthesia during the prodromal disease stage on age-dependent working memory decline in the symptomatic stage. Our results demonstrate that tDBS-nRE during the prodromal disease phase in young APP/PS1 mice prevents age-dependent working memory decline in the symptomatic stage. We strongly believe that these new results are of high importance and emphasize the role of thalamic nRE in both prodromal and symptomatic disease stages (Fig. 7).
- (2) We demonstrated that pDBS-nRE suppresses IESs exposed by several structurally distinct general anesthetics, irrespective of the specific type of the drug (Supplementary Fig. 5).
- (3) We provided additional analysis of IESs in mPFC, particularly focusing on their relationship to IESs in CA1 and their effect on working memory (Fig. 4).
- (4) We provided new data and computational analysis of transcriptomics in CA1 excitatory neurons using single-nuclei RNA sequencing (Fig. 1g-h and Supplementary Table 1) in 4 groups of mice (WT-Awake, WT-GA, fAD-Awake, fAD-GA). We identified molecular signatures of excitatory CA1 neurons which are specific for the anesthetized state in prodromal APP/PS1 mice (in collaboration with Naomi Habib lab).
- (5) We tested the effect of our general anesthesia protocol (1.5% isoflurane, 3 hr) on soluble A β 40 and A β 42 in the hippocampus (Supplementary Fig. 2).

Please see below our point-by-point reply (in blue) to the reviewers’ comments (in black).

Reviewer #1 (Remarks to the Author):

The studies by Shoob et al. revealed a neural circuit mechanism by which anesthesia dampens resilience to cognitive decline in Alzheimer's disease (AD). The works are significant by demonstrating the circuit mechanisms that underlie the resilience of AD cognitive impairment. More importantly, the studies used anesthesia as a clinically relevant tool to provide clinically relevant findings. The conclusion was supported by the data presented. The methodology was sound, and the data analysis was fine. However, there are some concerns.

1. The studies can be significantly improved by including biochemistry studies. Are there any changes of neurotransmission, e.g., NMDA, Tau, Abeta, or neuroinflammation, that are responsible for the changes in the neural circuit between nucleus reunions and hippocampus CA1 following anesthesia?

We would like to thank the reviewer for this question. We put forward a major effort and performed single-nuclei RNA sequencing (snRNAseq) of CA1 excitatory neurons (in collaboration with Naomi Habib lab) that are dysregulated by anesthesia in APP/PS1 mice¹. We analyzed the results of snRNAseq in 4 groups of mice (WT-Awake, WT-GA, fAD-Awake, and fAD-GA) which are presented in Fig. 1g-h and in Supplementary Table 1. The results point to 20 top differentially expressed genes related to synaptic function, neurodevelopment and neurodevelopmental disorders (in addition to genes related to proteostasis, neurodegeneration and sleep abnormalities). Moreover, we performed ELISA of soluble A β 40 and A β 42 peptides and found no significant changes by our anesthesia regime (3 hr of 1.5% isoflurane anesthesia, Supplementary Fig. 2). Here is the text in the revised manuscript:

“We next tested whether fAD mutations alter the transcriptional response of CA1 excitatory neurons (CA1-ExNs) to anesthesia, which could potentially shed light on the plasticity mechanisms associated with CA1-ExN dyshomeostasis¹ and memory deficits (Fig. 1e) in the model mice. To address this question, we utilized single nuclei RNA-sequencing (snRNAseq) to profile 5,796 nuclei of CA1-ExNs by and performed a comparative analysis based on brain state and genotype in 5-month-old mice. Hippocampi were isolated from WT and APP/PS1 littermates after wake or after general anesthesia (GA: 1.5% isoflurane, 3 hr), at the same circadian time. Four groups of mice (15 mice, 3-4 mice per group) were analyzed: WT-Awake, WT-GA, fAD-Awake, and fAD-GA. In awake state, we identified 89 differentially expressed genes (DEGs) between WT and APP/PS1 mice, while in the GA state, there were 119 DEGs (Fig. 1g and Supplementary Table 1). In both brain states (awake and GA), the DEGs were found to be enriched in pathways related to synapse organization, synaptic signaling, neural development, and neuronal projections. Notably, under anesthesia, additional DEGs between the genotypes were observed, which were related to RNA processing/splicing and cell motility (based on GO biological pathway enrichment analysis, average logFC > 0.2, FDR correction < 0.05). Among the top 20 DEGs under anesthesia (Fig. 1h), many are closely associated with synaptic functions, neurodevelopment and neurodevelopmental disorders²⁻¹¹ (e.g. *Nrg3os*, *Pigk*, *Ube3a*, *Kcnn2*, *Lrrc4*, *Fez1*, *Rsrp1*, *Aplp1*, *Ogt*, *Lin7b*). We also identified DEGs related to A β / tau proteostasis¹²⁻¹⁴ (*Rtn1*, *Dnaja2*, *Srsf7*), sleep abnormalities^{15,16} (*Gnas*, *Ube3a*), and neurodegeneration^{17,18} (*Ogt*, *Ssbp4*). As our anesthesia regime did not result in detectable changes in the levels of soluble A β 40 and A β 42 peptides or their ratio (Supplementary Fig. 2), it is unlikely that the observed transcriptional changes are downstream of A β .”

2. The pathogenesis of postoperative cognitive dysfunction is more complicated than anesthesia alone. More clinical studies suggest that anesthesia alone may not be associated with postoperative cognitive dysfunction. Thus, the authors need to tone down the conclusion

that the circuit between nRE and CA1 serves as the pathogenesis of postoperative dysfunction.

We agree with the reviewer that more clinical studies are needed to conclude whether on the causal role of anesthesia in POCD. In the revised manuscript, we modified the discussion: “However, whether the rate of IESs during anesthesia in presymptomatic AD patients can predict the severity of cognitive decline at later disease stages and whether cortico-hippocampal IESs play a role in POCD are still open questions that will require further investigation.”

3. Page 8, line 208: What is the purpose of the second session of general anesthesia?

In this experimental design, we used 2 sessions of general anesthesia: one before and another one after DBS to quantify epileptiform spike rate. The purpose of the second anesthesia in control group was to verify that the second anesthesia without DBS does not decrease the rate of epileptiform spikes by its own.

4. Page 9, line 236, the statement may not be relevant to the scope of the current studies. Please see comment two as well.

As we explained in our answer to the point #2, we fixed this problem.

5. Page 14, line 451-453: Why only male mice were used in behavioral studies, but both male and female mice were used in EP and calcium imaging studies?

The combination of electrophysiology and behavior in the same APP/PS1 mice are very complicated and labor-intensive. Therefore, we initially used only males to reduce the variability and the number of mice per group in order to reach the robust conclusions. In the new behavioral study that we conducted during the revision (Fig. 7), we included females as well. The student who performed behavioral experiment was blind to the electrophysiological data and perturbations.

Reviewer #2 (Remarks to the Author):

Review of Nature Communications: Shoob et a.

This elegant study of a neuronal system involved in AD pathophysiology and cognitive failures is interesting for a number of reasons. First of all, it is a system study of an AD model. While we often pay lip service to the idea that AD is a failure of neuronal systems, there are few studies of interactions between brain areas involved in the pathophysiology and cognitive symptoms of the disease that also study the mechanisms of those failures at the small neuronal assembly level. For example, studies relating cognitive decline to plaque load and tangles reveal that the relationship to abnormal protein accumulation and aggregation alone cannot completely account for the symptoms of the disease. Clearly, something more complex is involved, and interactions among neurons and brain area activity are a logical place to look (as first demonstrated many years ago by the Malinow lab in vitro). This is a point which should be emphasized in the discussion. Here we have an in vivo comparison of brain activity in the thalamic nRE-hippocampus-PFC system between a well-characterized APP mouse model and controls.

It is clear that AD pathophysiology is a complex matter: there are many papers on hyperexcitability of cortical neurons in APP model mice, but they fail to demonstrate convincing mechanisms, and intracellular recordings in vivo demonstrate that the hyperexcitability is limited to very restricted conditions depending on the state of the neuron. In this study, the authors demonstrate that hyperexcitability of hippocampal neurons is driven by inputs from the nRE and emerges during anesthesia, revealing that AD-model related hyperexcitability is not a simple matter of reduction of cortical inhibition, but is dependent on complex interactions among neurons in different areas. This paper is a timely study of disruption of system homeostasis as a mechanism involved in AD-model pathophysiology and cognitive decline.

We would like to thank the reviewer for encouraging comments. We made some modifications to discussion to emphasize these points.

The experiments themselves are well designed. I think that some additional analysis might be appropriate. For example, I would like to see the frequency spectra of the CA1 and PFC activity during the different sections of the experiment. This might shed some light on the specific mechanisms involved in the differences in levels of hyperactivity as well as providing information about levels of anesthesia. The Extended data would be a good place for such a figure.

We added PSD plots for LFP recordings in the CA1 (Supplementary Fig. 4e) and mPFC (Supplementary Fig. 6d) before/after pDBS-nRE under anesthesia. These plots don't add much information about IESs since they are dominated by the effects of anesthesia.

That being said, there are a few issues with the manuscript. One is that I found it difficult to read. It would be helpful to have small subtitles on the figure. For example, The title of Figure 2, is "...CA1 and mPFC hyperexcitability...", but I do not see any data relating to the PFC in that figure. If I am missing something, so will the other readers. In any case, such insets would greatly improve the readability of the study.

In any case, given that one of the major points is the interaction of the PFC with the other two brain areas, it would be nice to see some recording of those areas performed simultaneously

with those made in CA1, as shown in Figure 3 of the extended data. (I would say that that figure should be in the paper itself, as it is extremely important to the study.)

We would like to thank the reviewer for these suggestions. We moved the supplementary data related to the mPFC and Fig. 2l to the main Fig. 4 on the revised manuscript. We also performed additional cross-correlation analysis of IESs in CA1 and mPFC (Fig. 4c-f) as well as correlation between mPFC and behavior (Fig. 4h).

For me, the major issue with the manuscript is the fact that the only anesthetic used is isoflurane. Isoflurane has been shown to induce amyloid accumulation and apoptosis (Xie et al, J Neurosci. 2007 27:1247-1254), as well as affecting physiology of the cortical network in different ways than do other anesthetics. (See differences in EEG under isoflurane and halothane, for example). At the very least, the authors should address that issue in the discussion. However, in my opinion, if they could perform experiments similar to those in described in Figure 3 in the extended results, as well as figures 2 and 3 in the paper. under a different inhalant (or other) anesthetic, the study would be far more powerful. (I do realize that this may be difficult to do in a timely manner, as it may involve an addendum to animal protocols.) If this is not possible, it does not reduce the value of the results (as long as a qualifier is inserted to the effect that it may not be generalizable to all anesthetics; the study is important for the reasons stated above. However, if the effects of a second anesthetic are measured and compared, it would greatly increase the validity of the study.

We repeated the experiments under structurally different general anesthetics – ketamine+xylazine and added these results to the revised manuscript. Similarly to isoflurane, LFP recordings of the CA1 neuronal activity under ketamine-xylazine revealed frequent IESs in APP/PS1 mice (1.36 ± 0.17 per min, Supplementary Fig. 5a) that were effectively suppressed by tDBS-nRE (~50% decrease, Supplementary Fig. 5b). These findings indicate that the impact of tDBS-nRE is not specific to isoflurane-induced hyperexcitability.

Reviewer #3 (Remarks to the Author):

The manuscript by Shoob and colleagues describe a set of experiments linking epileptiform spikes recorded in the hippocampus under anesthesia in prodromal AD-model mice to aberrant regulation of input from the nucleus reunions of the thalamus. The authors employ a multi-disciplinary approach to convincingly demonstrate that dysregulation in this circuit, unmasked by anesthesia, contributes to deficits in spatial working memory observed post-anesthesia in AD model, but not WT, mice. Overall the experiments are well thought out and executed, and I have only minor technical comments. My chief criticism for the work presented stems from the difficulty understanding the "big-picture" question that is actually addressed by the studies.

The logic presented in the abstract and introduction suggests that the authors believe the results applicable to resilience against cognitive decline in AD. What is being assessed, however, is a neuro-behavioral phenotype observed prior to the onset of cognitive impairments known to exist in this particular humanized APP mouse line. Independent of known problems with the amyloid-beta hypothesis of AD, properly speaking this would make it a study of predictors of cognitive decline in prodromal disease phases. A question that flows naturally from this would be whether or not anesthesia is indeed unmasking a mechanism of cognitive impairment that is relevant to later-stage disease?

We would like to thank the reviewer for asking this important question. We completely agree that understanding whether or not anesthesia is indeed unmasking a mechanism of cognitive impairment that is relevant to symptomatic stage disease is central to our study. In order to address this question, we (1) tested if a reduction in epileptiform activity during the prodromal phase by tDBS-nRE prevents age-dependent cognitive decline at later stage; (2) analyzed the correlation between the rate of epileptiform spikes during anesthesia in the prodromal phase to the degree of cognitive decline later on (during symptomatic phase). We invested large efforts to conduct a longitudinal study in which the tonic stimulation of nRE was started in young mice (4-5 m.o., prodromal stage) and their cognitive performance was tested at later stage in 8-9 m.o. APP/PS1 mice, displaying age-dependent working memory decline. The tDBS-nRE stimulation was repeated monthly until the training and the testing of working memory have been performed. The behavioral tests were conducted blindly, by the person who was not involved in DBS and electrophysiological recordings. We are very excited by our new data suggesting that tDBS-nRE in 'young' APP/PS1 mice prevents age-dependent decline in spatial working memory at later disease stage. Furthermore, we show an inverse correlation between the rate of CA1 ESs in young mice and working memory in the same old APP/PS1 mice. These new results are summarized in the new Fig. 7 of the revised manuscript.

It would seem important then to demonstrate that there is higher epileptiform activity in older, cognitively impaired mice. Would we then suspect that these impairments would also be reversed by the tonic stimulation protocol they describe?

The disease might have a more complicated and complex progression. CA1 IESs decline with age in APP/PS1 mice¹⁹. We propose that early hyperexcitability exposed by low-arousal states such as sleep and anesthesia induces maladaptive plasticity leading to cognitive decline. Follow-up study is needed to test whether tDBS-nRE during the symptomatic stage can reverse age-dependent memory decline.

It also seems important to identify, mechanistically, what exactly the anesthesia unveiling. What underlies the ability of isoflurane to uniquely tax this nRE-CA1 circuit? Is the effect specific to ISO? What are potential neuropharmacological mechanisms?

To address these questions, we repeated the experiments under structurally different general anesthetics – ketamine+xylazine and added these results to the revised manuscript. Similarly to isoflurane, LFP recordings of the CA1 neuronal activity under ketamine-xylazine revealed frequent IESs in APP/PS1 mice (1.36 ± 0.17 per min, Supplementary Fig. 5a) that were effectively suppressed by tDBS-nRE (~50% decrease, Supplementary Fig. 5b). These findings indicate that the impact of tDBS-nRE is not specific to isoflurane-induced hyperexcitability.

We also put forward a major effort and performed single-nuclei RNA sequencing (snRNAseq) of CA1 excitatory neurons (in collaboration with Naomi Habib lab) that are dysregulated by anesthesia in APP/PS1 mice¹. The results of snRNAseq in 4 groups of mice (WT-Awake, WT-GA, fAD-Awake, and fAD-GA) are summarized on pages 4-5 of the revisited manuscript (Fig. 1g-h and in Supplementary Table 1). Among the top 20 DEGs under anesthesia, many are closely associated with synaptic functions, neurodevelopment and neurodevelopmental disorders²⁻¹¹ (e.g. *Nrg3os*, *Pigk*, *Ube3a*, *Kcnn2*, *Lrrc4*, *Fez1*, *Rsrp1*, *Aplp1*, *Ogt*, *Lin7b*). We also identified DEGs related to A β / tau proteostasis¹²⁻¹⁴ (*Rtn1*, *Dnaja2*, *Srsf7*), sleep abnormalities^{15,16} (*Gnas*, *Ube3a*), and neurodegeneration^{17,18} (*Ogt*, *Ssbp4*). As our anesthesia regime did not result in detectable changes in the levels of soluble A β 40 and A β 42 peptides or their ratio (Supplementary Fig. 2), it is unlikely that the observed transcriptional changes are downstream of A β .

With respect to this as a model of the vulnerability of prodromal AD patients to the negative side effects of anesthesia, it would seem important to identify the long term implications, e.g. do mice exposed to anesthesia in the prodromal phase have greater impairment later?

This is indeed an important and interesting question that was not the focus of the current study. Future follow-up studies are required to test many parameters that can affect long-term effects, such as the length, the number of repeats and timing of the anesthesia sessions.

Are you tapping into a circuit that later contributes to cognitive impairments, or this specific to the prodromal phase?

We believe that based on our new results summarized in the Fig. 7, we can conclude that nRE-CA1-mPFC circuitry is involved in both, prodromal and symptomatic disease phases.

It also seems important to determine if these same circuit and behavioral effects are specific to the AD context, or if the same is observed in aged mice, where cognition declines and vulnerability to the negative effects of anesthesia increases.

These are all intriguing questions that are beyond the scope of this project. Answering these questions would require a separate project in aged mice that would take several years to complete.

If ES spikes are the mechanism underlying SWM deficits, can you induce them in a WT mouse and create SWM deficits?

This is an excellent question for future studies. Currently, we are submitting a grant and ethical protocol for this follow-up project.

In all, the manuscript as presented struggles from a bit of an identity crisis, which ultimately limits its overall impact.

We believe that our new experiments added to the revised manuscript solve the ‘identity crisis’ identified by the reviewer and provide the evidence on the role of nRE-CA1-mPFC circuitry in both prodromal and symptomatic phases of the disease. We thank the reviewer for his suggestions and believe that these new experiments strengthened the revised manuscript.

Minor points:

Ca imaging: What was actually quantified here? How many cells per animal? Most of this appears to be ANOVAs run on all cells from all mice. Begs the question of how consistent this is across mice. A more appropriate analysis would be something like a linear mixed effect model with mouse and imaging session as random effects. For the accompanying figure, this data really needs to be shown side by side with the data from WT mice.

In these experiments, we plotted the average distribution of spike rates (inferred from Ca imaging) per brain state across 3 mice and sessions (Fig. 5g-l in the revised manuscript). The sessions are awake/anesthetized 1 day before tDBS, 1 week after tDBS and 1 month after tDBS in matching order). The number of cells per mouse are: 1586, 1817 and 1326. We quantified the total activity, defined as the number of active neurons multiplied by mean firing rate and plotted the % of total activity in anesthetized state from awake state, per session (same mouse, same day, miniscope was not removed or adjusted between recording sessions of the 2 states) for 3 separate sessions in each mouse (Fig. 5f). We used a grouped analysis (Kruskal-Wallis test with mouse and imaging session as random effects). These results show robustness of the data. The supplementary figure extends it to all the time points in comparison to WT.

For the tonic stimulation: why use this as opposed to DREADD-based inhibition. The latter is a little easier to understand mechanistically, could be useful for contextualizing what the tonic DBS is doing.

There are several reasons we chose the DBS approach for the SWM experiments:

1. tDBS-nRE had a long lasting effect, rescuing activity homeostasis even 1 month after the stimulation (Fig. 5). In contrast, CNO effect was lasting only few hours, matching its half-life. This makes DBS approach is more attractive.
2. Currently, DBS approach is more clinically relevant. It will take some time until safe AAV therapy will be developed to target specific brain areas with mutant muscarinic receptors.

Relationship between mPFC ES and SWM? Detail the relationship between mPFC ES and SWM, how does it overlap with hippocampus ES relationship to behavior, where does it differ?

We added the correlation between success rate in the delta maze and the rate of IESs in the mPFC to the manuscript (Fig. 4h, Spearman $r = -0,61$, $P < 0.05$). In addition, we performed new analysis of IESs in the mPFC and their relation to IESs in the CA1 (Fig. 4c-g).

References:

- 1 Zarhin, D. *et al.* Disrupted neural correlates of anesthesia and sleep reveal early circuit dysfunctions in Alzheimer models. *Cell Rep* **38**, 110268 (2022). <https://doi.org:10.1016/j.celrep.2021.110268>
- 2 Kim, S. *et al.* NGL family PSD-95–interacting adhesion molecules regulate excitatory synapse formation. *Nature Neuroscience* **9**, 1294-1301 (2006). <https://doi.org:10.1038/nn1763>
- 3 Ou, G. Y., Lin, W. W. & Zhao, W. J. Neuregulins in Neurodegenerative Diseases. *Front Aging Neurosci* **13**, 662474 (2021). <https://doi.org:10.3389/fnagi.2021.662474>

- 4 Smith, S. E. P. *et al.* Increased Gene Dosage of Ube3a Results in Autism Traits and Decreased Glutamate Synaptic Transmission in Mice. *Science translational medicine* **3**, 103ra197-103ra197 (2011). <https://doi.org/doi:10.1126/scitranslmed.3002627>
- 5 Zucker, B. *et al.* Decreased Lin7b expression in layer 5 pyramidal neurons may contribute to impaired corticostriatal connectivity in huntington disease. *J Neuropathol Exp Neurol* **69**, 880-895 (2010). <https://doi.org/10.1097/NEN.0b013e3181ed7a41>
- 6 Lagerlöf, O., Hart, G. W. & Haganir, R. L. O-GlcNAc transferase regulates excitatory synapse maturity. *Proceedings of the National Academy of Sciences* **114**, 1684-1689 (2017). <https://doi.org/doi:10.1073/pnas.1621367114>
- 7 Schilling, S. *et al.* APLP1 Is a Synaptic Cell Adhesion Molecule, Supporting Maintenance of Dendritic Spines and Basal Synaptic Transmission. *J Neurosci* **37**, 5345-5365 (2017). <https://doi.org/10.1523/jneurosci.1875-16.2017>
- 8 Mochel, F. *et al.* Variants in the SK2 channel gene (KCNN2) lead to dominant neurodevelopmental movement disorders. *Brain : a journal of neurology* **143**, 3564-3573 (2020). <https://doi.org/10.1093/brain/awaa346>
- 9 Nguyen, T. T. M. *et al.* Bi-allelic Variants in the GPI Transamidase Subunit PIGK Cause a Neurodevelopmental Syndrome with Hypotonia, Cerebellar Atrophy, and Epilepsy. *The American Journal of Human Genetics* **106**, 484-495 (2020). [https://doi.org:https://doi.org/10.1016/j.ajhg.2020.03.001](https://doi.org/https://doi.org/10.1016/j.ajhg.2020.03.001)
- 10 Kang, E. *et al.* Interaction between FEZ1 and DISC1 in Regulation of Neuronal Development and Risk for Schizophrenia. *Neuron* **72**, 559-571 (2011). [https://doi.org:https://doi.org/10.1016/j.neuron.2011.09.032](https://doi.org/https://doi.org/10.1016/j.neuron.2011.09.032)
- 11 Osenberg, S. *et al.* Activity-dependent aberrations in gene expression and alternative splicing in a mouse model of Rett syndrome. *Proceedings of the National Academy of Sciences* **115**, E5363-E5372 (2018). <https://doi.org/doi:10.1073/pnas.1722546115>
- 12 He, W. *et al.* Reticulon family members modulate BACE1 activity and amyloid- β peptide generation. *Nature Medicine* **10**, 959-965 (2004). <https://doi.org/10.1038/nm1088>
- 13 Gao, L., Wang, J., Wang, Y. & Andreadis, A. SR protein 9G8 modulates splicing of tau exon 10 via its proximal downstream intron, a clustering region for frontotemporal dementia mutations. *Molecular and Cellular Neuroscience* **34**, 48-58 (2007).
- 14 Mok, S.-A. *et al.* Mapping interactions with the chaperone network reveals factors that protect against tau aggregation. *Nature structural & molecular biology* **25**, 384-393 (2018). <https://doi.org/10.1038/s41594-018-0057-1>
- 15 Lassi, G. *et al.* Loss of Gnas Imprinting Differentially Affects REM/NREM Sleep and Cognition in Mice. *PLOS Genetics* **8**, e1002706 (2012). <https://doi.org/10.1371/journal.pgen.1002706>
- 16 Colas, D., Wagstaff, J., Fort, P., Salvert, D. & Sarda, N. Sleep disturbances in Ube3a maternal-deficient mice modeling Angelman syndrome. *Neurobiology of Disease* **20**, 471-478 (2005). [https://doi.org:https://doi.org/10.1016/j.nbd.2005.04.003](https://doi.org/https://doi.org/10.1016/j.nbd.2005.04.003)
- 17 Wang, A. C., Jensen, E. H., Rexach, J. E., Vinters, H. V. & Hsieh-Wilson, L. C. Loss of O-GlcNAc glycosylation in forebrain excitatory neurons induces neurodegeneration. *Proceedings of the National Academy of Sciences* **113**, 15120-15125 (2016). <https://doi.org/doi:10.1073/pnas.1606899113>
- 18 Yokoyama, J. S. *et al.* Association Between Genetic Traits for Immune-Mediated Diseases and Alzheimer Disease. *JAMA Neurol* **73**, 691-697 (2016). <https://doi.org/10.1001/jamaneurol.2016.0150>
- 19 Soula, M. *et al.* Interictal epileptiform discharges affect memory in an Alzheimer's Disease mouse model. *bioRxiv* (2023). <https://doi.org/10.1101/2023.02.15.528683>

REVIEWER COMMENTS

Reviewer #1 (Remarks to the Author):

The authors should be congratulated for nicely addressing the comments raised from the original submission. The quality of the manuscript has been significantly improved by the revision. This reviewer has no additional comments.

Reviewer #2 (Remarks to the Author):

The authors have made great efforts to address the concerns of the reviewers. I am satisfied with their responses to my criticisms, and especially appreciate the additional experiments performed under different anesthetics. These are not easy experiments, and go a long way to enhancing the validity of the study.

Prof. Edward A. Stern

Reviewer #3 (Remarks to the Author):

The revised manuscript by Shoob et al has been strengthened by the addition of new data. In general, I think the new manuscript is more impactful, and I appreciate the emphasis on abnormalities in neurophysiology in low-arousal states in the revised introduction. I recommend sticking to this argument throughout, as opposed framing as an investigation into the causal link between anesthesia and dementia (in the second paragraph of page 1). The finding of the robust, long-term effect on nRE tonic stimulation on hippocampal physiology and cognitive impairment is strong, and may have clinical implications. My comments below are meant to point out areas that somewhat detract from this major result.

1). Multiple reviewers noted that the manuscript would be enhanced by including another anesthetic. The author's did use ketamine to demonstrate similiar effects of tonic stimulation on the reduction of IESs, though it's difficult to directly compare the two, indeed, figure 3F would suggest the rate of CA1 IESs is higher under ISO than ketamine (scales are rather different in 3H, I compared to Figure S5). Indeed CNO treatment appears to have reduced IES frequency to a level similar to that observed with ketamine at baseline. Begs the question in the behavioral effect of ketamine is similiar to Isoflurane, and if not, does it relate to the general differences in the physiological effect of ketamine and ISO.

2). It is well known that there are sex differences in the pathology of fAD mouse lines. Generally speaking, the female mice appear to have higher levels of central and functional impairment. It is unfortunate that the behavioral tests in these mice were performed in males, yet the physiological/genetic endpoints employ both sexes. The authors would ideally power the study to look at each sex, but at minimum should provide justification for the use of single or both sexes across experiments. The N per sex in each experiment should also be included.

3). The DREADD manipulation provides an easily interpret-able test of mechanism (e.g. increased nRE drive on CA1), however, the experiment appears to have been conducted within-subject, comparing systemic CNO (5mg/kg) to vehicle. Whether the study design was within or between should be clarified, but either way CNO is converted to clozapine in vivo, and 5 mg/kg is a relatively high dose. Granted, the impact on the physiology of the circuit is the focus, and is quantified, but controls for the potential impact of systemic, low dose clozapine on their physiological endpoints would facilitate interpretation.

Reviewer #1 (Remarks to the Author):

The authors should be congratulated for nicely addressing the comments raised from the original submission. The quality of the manuscript has been significantly improved by the revision. This reviewer has no additional comments.

Reviewer #2 (Remarks to the Author):

The authors have made great efforts to address the concerns of the reviewers. I am satisfied with their responses to my criticisms, and especially appreciate the additional experiments performed under different anesthetics. These are not easy experiments, and go a long way to enhancing the validity of the study.

Prof. Edward A. Stern

We would like to thank both reviewers for their appreciation of our work.

Reviewer #3 (Remarks to the Author):

The revised manuscript by Shoob et al has been strengthened by the addition of new data. In general, I think the new manuscript is more impactful, and I appreciate the emphasis on abnormalities in neurophysiology in low-arousal states in the revised introduction. I recommend sticking to this argument throughout, as opposed framing as an investigation into the causal link between anesthesia and dementia (in the second paragraph of page 1). The finding of the robust, long-term effect on nRE tonic stimulation on hippocampal physiology and cognitive impairment is strong, and may have clinical implications. My comments below are meant to point out areas that somewhat detract from this major result.

We would like to thank the reviewer for finding our findings to be robust, impactful, and clinically relevant. We agree with the reviewer's suggestion to emphasize the abnormalities in low-arousal states and changed the second paragraph of the introduction accordingly.

1). Multiple reviewers noted that the manuscript would be enhanced by including another anesthetic. The author's did use ketamine to demonstrate similar effects of tonic stimulation on the reduction of IESs, though it's difficult to directly compare the two, indeed, figure 3F would suggest the rate of CA1 IESs is higher under ISO than ketamine (scales are rather different in 3H, I compared to Figure S5). Indeed CNO treatment appears to have reduced IES frequency to a level similar to that observed with ketamine at baseline. Begs the question in the behavioral effect of ketamine is similar to Isoflurane, and if not, does it relate to the general differences in the physiological effect of ketamine and ISO.

We appreciate the thoughtful comments of the reviewer regarding the lower IES rate under ketamine-based anesthesia. We have carefully considered the potential inclusion of ketamine experiments and have come to the conclusion that they may not significantly contribute to our study's objectives. Here's our rationale: If ketamine experiments yield negative results, (e.g., ketamine anesthesia does not impair working memory), they might be attributed to the lower IES rate, and therefore, wouldn't contradict our conclusions based on isoflurane experiments. If the results are positive (e.g., ketamine anesthesia impairs working memory), they won't add much due to the following reasons:

Ketamine is already widely recognized for its disruptive effects on working memory across species, including humans¹⁻⁵, primates⁶⁻⁸ and wild-type rodents⁹⁻¹². These disruptive effects of ketamine are evident across a wide range of concentrations, including anesthetic and sub-

anesthetic doses and are primarily mediated by the blockade of NMDARs¹³. In addition, a single injection of anesthetic dose of ketamine produces anxiety-like state¹⁴ and chronic sub-anesthetic dose of ketamine induces schizophrenia-like state¹⁵. Our preliminary data align with existing literature, demonstrating that ketamine-xylazine anesthesia, administered at the same dosage and duration as used in our paper, impairs contextual fear memory in WT mice. Given this established effect of ketamine on memory functions, irrespective of IESs, we believe that behavioral ketamine experiments may not be conducive to our study's goals.

It is important to note that our study's conclusions regarding the causal link between IES rate and working memory extend beyond anesthesia. As illustrated in our new experiment added during the revision (Fig. 7), we have demonstrated the effect of tDBS on age-dependent working memory impairments is independent of anesthesia exposure. Therefore, we do not believe that the effect of IES rate on working memory requires anesthesia. Additional experiments involving ketamine may not significantly enhance the outcome but will produce long delay in publication and are against ethical considerations to prevent the use of a large number of animals.

2). It is well known that there are sex differences in the pathology of fAD mouse lines. Generally speaking, the female mice appear to have higher levels of central and functional impairment. It is unfortunate that the behavioral tests in these mice were performed in males, yet the physiological/genetic endpoints employ both sexes. The authors would ideally power the study to look at each sex, but at minimum should provide justification for the use of single or both sexes across experiments. The N per sex in each experiment should also be included.

When we initially embarked on this project, the effectiveness of DBS-nRE was unknown. The experiments involved are very demanding and require a specialized skill set. Specifically, they necessitate the expertise of an experienced electrophysiologist capable of verifying the precise positioning of the stimulating electrode in the nRE by recording monosynaptic nRE-CA1 synaptic transmission. Additionally, conducting longitudinal CA1 / mPFC recordings in conjunction with behavioral assessments (taking at least month of training before even starting the assessment of working memory at different delays), all while ensuring the experimenter remains blind to electrophysiological manipulations, adds to the complexity.

Given both the intricacies of this experiment and the substantial body of previous behavioral studies conducted by in our lab and others', we opted to focus on male APP/PS1 mice. This decision aimed to reduce variability and the number of mice required to achieve robust conclusions in our study. However, we agree that follow-up studies should incorporate females to ensure a more comprehensive understanding of the potential gender-related effects.

We added the numbers of females and males used for electrophysiological recordings to the figures' legends. We also added Figure S10 showing that the physiological parameters and the rate of IESs under anesthesia is similar between male and female APP/PS1 mice.

3). The DREADD manipulation provides an easily interpret-able test of mechanism (e.g. increased nRE drive on CA1), however, the experiment appears to have been conducted within-subject, comparing systemic CNO (5mg/kg) to vehicle. Whether the study design was within or between should be clarified, but either way CNO is converted to clozapine in vivo, and 5 mg/kg is a relatively high dose. Granted, the impact on the physiology of the circuit is the focus, and is quantified, but controls for the potential impact of systemic, low dose clozapine on their physiological endpoints would facilitate interpretation.

We greatly appreciate the reviewer's suggestion regarding the addition of a control group in our chemogenetic experiments. We included a control group of APP/PS1 mice that were infected with Cre-dependent mCherry in the nRE and AAV2retro-CaMKIIa-iCre in the CA1.

We subsequently measured the effect of CNO on IES rate in this group of mice that does not express Gi-DREADD to provide a clear baseline for comparison.

We believe that these additional data and explanations convincingly address the reviewer's concerns.

References:

- 1 Zhornitsky, S. *et al.* Acute effects of ketamine and esketamine on cognition in healthy subjects: A meta-analysis. *Progress in Neuro-Psychopharmacology and Biological Psychiatry* **118**, 110575 (2022). [https://doi.org:https://doi.org/10.1016/j.pnpbp.2022.110575](https://doi.org/https://doi.org/10.1016/j.pnpbp.2022.110575)
- 2 Ghoneim, M. M., Hinrichs, J. V., Mewaldt, S. P. & Petersen, R. C. Ketamine: behavioral effects of subanesthetic doses. *J Clin Psychopharmacol* **5**, 70-77 (1985).
- 3 Adler, C. M., Goldberg, T. E., Malhotra, A. K., Pickar, D. & Breier, A. Effects of ketamine on thought disorder, working memory, and semantic memory in healthy volunteers. *Biol Psychiatry* **43**, 811-816 (1998). [https://doi.org:10.1016/s0006-3223\(97\)00556-8](https://doi.org/10.1016/s0006-3223(97)00556-8)
- 4 Morgan, C. J., Mofeez, A., Brandner, B., Bromley, L. & Curran, H. V. Acute effects of ketamine on memory systems and psychotic symptoms in healthy volunteers. *Neuropsychopharmacology* **29**, 208-218 (2004). [https://doi.org:10.1038/sj.npp.1300342](https://doi.org/10.1038/sj.npp.1300342)
- 5 Rybakowski, J. K. *et al.* Ketamine Anesthesia, Efficacy of Electroconvulsive Therapy, and Cognitive Functions in Treatment-Resistant Depression. *The Journal of ECT* **32**, 164-168 (2016). [https://doi.org:10.1097/yct.0000000000000317](https://doi.org/10.1097/yct.0000000000000317)
- 6 Roussy, M. *et al.* Ketamine disrupts naturalistic coding of working memory in primate lateral prefrontal cortex networks. *Mol Psychiatry* **26**, 6688-6703 (2021). [https://doi.org:10.1038/s41380-021-01082-5](https://doi.org/10.1038/s41380-021-01082-5)
- 7 Nakako, T. *et al.* Effects of a dopamine D1 agonist on ketamine-induced spatial working memory dysfunction in common marmosets. *Behavioural Brain Research* **249**, 109-115 (2013). [https://doi.org:https://doi.org/10.1016/j.bbr.2013.04.012](https://doi.org/https://doi.org/10.1016/j.bbr.2013.04.012)
- 8 Paule, M. G. *et al.* Ketamine anesthesia during the first week of life can cause long-lasting cognitive deficits in rhesus monkeys. *Neurotoxicology and teratology* **33** **2**, 220-230 (2011).
- 9 Goswamee, P. *et al.* Effects of subanesthetic ketamine and (2R,6R) hydroxynorketamine on working memory and synaptic transmission in the nucleus reuniens in mice. *Neuropharmacology* **208**, 108965 (2022). <https://doi.org:https://doi.org/10.1016/j.neuropharm.2022.108965>
- 10 Pitsikas, N. & Boultaadakis, A. Pre-training administration of anesthetic ketamine differentially affects rats' spatial and non-spatial recognition memory. *Neuropharmacology* **57**, 1-7 (2009). <https://doi.org:https://doi.org/10.1016/j.neuropharm.2009.03015>.
- 11 Wang, J. H., Fu, Y., Wilson, F. A. W. & Ma, Y. Y. Ketamine affects memory consolidation: Differential effects in T-maze and passive avoidance paradigms in mice. *Neuroscience* **140**, 993-1002 (2006). <https://doi.org:https://doi.org/10.1016/j.neuroscience.2006.02.062>
- 12 Sun, Y. T. *et al.* Effect of ketamine anesthesia on cognitive function and immune function in young rats. *Cell Mol Biol (Noisy-le-grand)* **62**, 63-66 (2016).
- 13 Lisman, J. E., Fellous, J.-M. & Wang, X.-J. A role for NMDA-receptor channels in working memory. *Nature Neuroscience* **1**, 273-275 (1998). [https://doi.org:10.1038/1086](https://doi.org/10.1038/1086)

- 14 Pitsikas, N., Georgiadou, G., Delis, F. & Antoniou, K. Effects of Anesthetic Ketamine on Anxiety-Like Behaviour in Rats. *Neurochemical Research* **44**, 829-838 (2019). <https://doi.org:10.1007/s11064-018-02715-y>
- 15 Frohlich, J. & Van Horn, J. D. Reviewing the ketamine model for schizophrenia. *J Psychopharmacol* **28**, 287-302 (2014). <https://doi.org:10.1177/0269881113512909>

REVIEWERS' COMMENTS

Reviewer #3 (Remarks to the Author):

The authors additional data has largely addressed my previous concerns. The CNO control data helps a lot. The point regarding the ketamine comparison is taken, and think it is worth addressing this issue in the discussion (as presented in the letter). Similiarly, I appreciate the inclusion of the the n per mouse sex, and would only recommend that an acknowledgment of the biological sex differences in CNS and behavioral phenotypes as well as importance of further pursuing this, be included in the discussion of the results (as presented in the rebuttal letter).

Otherwise, I have no further substantive concerns, and the authors should be congratulated for their massive effort.

Reviewer #3 (Remarks to the Author):

The authors additional data has largely addressed my previous concerns. The CNO control data helps a lot. The point regarding the ketamine comparison is taken, and think it is worth addressing this issue in the discussion (as presented in the letter). Similiarly, I appreciate the inclusion of the the n per mouse sex, and would only recommend that an acknowledgment of the biological sex differences in CNS and behavioral phenotypes as well as importance of further pursuing this, be included in the discussion of the results (as presented in the rebuttal letter).

Otherwise, I have no further substantive concerns, and the authors should be congratulated for their massive effort.

We would like to thank the reviewer for his/her appreciation of our work. According to the reviewer's suggestions, we added the following sentences to the paper (emphasized by red font):

Page. 8: “Given that ketamine impairs working memory function irrespective of IESs even in wild-type rodents^{63,64} via NMDA receptor blockade⁶⁵, we did not continue investigating its effects on working memory in fAD mice.”

Page 12: “As our present study exclusively involved behavioral experiments in male mice, future research is needed to address the efficacy of tDBS-nRE in female model mice.”